# Domain-Robust Visual Imitation Learning with Mutual Information Constraints

**Edoardo Cetin & Oya Celiktutan**
Centre for Robotics Research
Department of Engineering, King's College London
{edoardo.cetin,oya.celiktutan}@kcl.ac.uk

## Abstract

Human beings are able to understand objectives and learn by simply observing others perform a task. Imitation learning methods aim to replicate such capabilities, however, they generally depend on access to a full set of optimal states and actions taken with the agent's actuators and from the agent's point of view. In this paper, we introduce a new algorithm – called *Disentangling Generative Adversarial Imitation Learning (DisentanGAIL)* – with the purpose of bypassing such constraints. Our algorithm enables autonomous agents to learn directly from high dimensional observations of an expert performing a task, by making use of adversarial learning with a latent representation inside the discriminator network. Such latent representation is regularized through mutual information constraints to incentivize learning only features that encode information about the completion levels of the task being demonstrated. This allows to obtain a shared feature space to successfully perform imitation while disregarding the differences between the expert's and the agent's domains. Empirically, our algorithm is able to efficiently imitate in a diverse range of control problems including balancing, manipulation and locomotive tasks, while being robust to various domain differences in terms of both environment appearance and agent embodiment.

## 1 Introduction

Recent advances demonstrated the strengths of combining reinforcement learning (RL) with powerful function approximators to obtain effective behavior for high dimensional control tasks (Lillicrap et al., 2015; Schulman et al., 2017; Haarnoja et al., 2018a). However, RL's reliance on a reward function introduces a fundamental limitation as reward specification and instrumentation can bring about a great design burden to potential users aiming to train an agent for a novel problem.

An alternative approach for addressing this limitation is to recover a learning signal through expert demonstrations. Most of the past work exploring this area focused on the problem setting where demonstrations are provided directly from the agent's point of view and through the agent's actuators, which we refer to as *agent-centric* imitation. However, applying *agent-centric* imitation for real-world robot learning would demand users to provide a diverse range of kinesthetic or teleoperated demonstrations to a robotic platform, leading to an unnatural user-agent interaction process.

In this paper, we focus instead on learning effective policies solely from a set of *external*, high dimensional observations of a different expert agent executing a task. We refer to this problem formulation as *observational* imitation. Solving this requires disentangling the expert's intentions from the observations' context, which has been a challenging problem for prior research, and often relied on additional assumptions about the environment and expert data (Torabi et al., 2019).

We propose a novel algorithm, called *Disentangling Generative Adversarial Imitation Learning (DisentanGAIL)*, to acquire effective agent behavior without such limitations. Our technique is based on the framework of inverse reinforcement learning, yet, it enables an agent to learn with only access to observations collected by watching a structurally different expert. *DisentanGAIL* utilizes an off-policy learner alongside a novel discriminator with a latent representation bottleneck, regularized to represent a domain invariant space over the agent's and expert's sets of observations.

This is achieved by enforcing two constraints on the estimated mutual information between the latent representation and the origin of collected observations.

In particular, the contribution of this work for solving *observational* imitation is threefold:

- We propose a discriminator making use of novel mutual information constraints, and provide techniques to adaptively and consistently ensure their enforcement.
- We identify the problem of *domain information disguising* when estimating mutual information and propose structural modifications to our models for its prevention.
- We show that, unlike prior work, our algorithm can scale to high dimensional tasks while being robust to domain differences in both environment appearance and agent embodiment, by testing on a novel diverse set of tasks with varying difficulty.

## 2 RELATED WORK

*Agent-centric* imitation has been a long-studied problem setting. *Behavior cloning* (Pomerleau, 1989; 1991; Ross et al., 2011) was first proposed to approach imitation from a supervised learning perspective. Particularly, an agent is trained to maximize the likelihood of executing a set of recorded optimal actions from the states encountered by the expert. *Inverse reinforcement learning* (IRL) (Ng et al., 2000; Abbeel & Ng, 2004; Ratliff et al., 2006) was more recently proposed as an alternative two-step solution to imitation and has been often shown to be more effective. First, IRL aims to recover a reward function by parameterizing a discriminator, trained to be representative of the objective portrayed by the expert demonstrations. Second, it tries to learn behavior to accomplish such objective, utilizing RL. To effectively understand and represent the expert's intentions, modern instantiations of IRL combined the maximum entropy problem formulation (Ziebart et al., 2008) with deep learning (Wulfmeier et al., 2015; Finn et al., 2016b), and proposed a direct connection with adversarial learning (Ho & Ermon, 2016; Finn et al., 2016a; Kostrikov et al., 2018). This allowed for successful imitation in complex control tasks with few expert demonstrations. Related to our algorithm, Peng et al. (2018) implemented a variational bottleneck to limit information flow in the discriminator, for tackling the discriminator saturation problem (Arjovsky & Bottou, 2017). Additionally, Zolna et al. (2019) proposed to optimize the discriminator to be maximally uncertain about uninformative sets of data as a form of regularization to disregard irrelevant features.

A different line of research instead considered imitating from observations of an expert performing a task, in problem settings resembling *observational* imitation. Earlier methods proposed to use hand-engineered mappings to domain invariant features (Gioioso et al., 2012; Gupta et al., 2016), while more recent works proposed to learn such mappings, relying on specific prior data obtained under both the agent and expert perspectives. These techniques include using time-aligned demonstrations (Gupta et al., 2017; Sermanet et al., 2018; Liu et al., 2018; Sharma et al., 2019), or multiple tasks where the agent and the expert already achieved expertise (Smith et al., 2019; Kim et al., 2019).

While effective, these methods for *observational* imitation make considerable assumptions about the task structure and the available data. Therefore, they have limited applicability for arbitrary problems, where environment instrumentation and prior knowledge are minimal. The work most related to ours is by Stadie et al. (2017), where a domain invariant representation is learned through utilizing a domain confusion loss that requires two different expert policies for sampling failure and success demonstrations in the expert domain. While this approach was also adopted in recent works (Okumura et al., 2020; Choi et al., 2020), empirically, it yielded successful imitation results only when working in low dimensional control tasks and with the agent and expert domains differing solely in their appearance. On the contrary, our algorithm only requires a limited set of expert demonstrations and allows for successful imitation in both low and high dimensional control tasks, with the expert's and agent's domains differing in both environment appearance and agent embodiment.

## 3 BACKGROUND AND PRELIMINARIES

### 3.1 ADVERSARIAL IMITATION LEARNING

In imitation learning, the agent is provided a set of expert demonstrations $B_E = \{\tau_1, \tau_2, ..., \tau_N\}$, where each $\tau = (s_0, a_0, s_1, a_1, ..., s_T)$ represents a trajectory collected with an expert policy from

the agent's point of view. These demonstrations are used to provide the learning signal for the agent to improve its policy $\pi$. In Generative Adversarial Imitation Learning (GAIL) (Ho & Ermon, 2016) this learning signal is obtained through a pseudo-reward function $R_D$, derived from a discriminator network $D$ trained to discern between 'expert' and 'agent' *state-action-next state* triplets:

$$\arg\max_D \mathbb{E}_{B_E} \log(D(s_i, a_i, s_{i+1})) + \mathbb{E}_{p_\pi(\tau)} \log(1 - D(s_i, a_i, s_{i+1})), \tag{1}$$

where $p_\pi(\tau)$ is the distribution of trajectories encountered by the agent, stemming from both the environment's dynamics and its own current policy $\pi$. Reinforcement learning methods are then applied for $\pi$ to adversarially maximize the sum of encountered pseudo-rewards:

$$\arg\max_\pi \mathbb{E}_{p_\pi(\tau)} \left[ \sum_{t=0}^{T-1} R_D(s_t, a_t, s_{t+1}) \right], \tag{2}$$

where $R_D(s_i, a_i, s_{i+1}) = \log(D(s_i, a_i, s_{i+1})) - \log(1 - D(s_i, a_i, s_{i+1}))$. Ho & Ermon (2016) proposed to iteratively execute the adversarial optimization steps described in Eqs. 1 and 2, optimizing $\pi$ through the Trust Region Policy Optimization algorithm (Schulman et al., 2015).

## 3.2 *Observational* IMITATION LEARNING

In the problem setting of *observational* imitation, we are concerned with learning without knowledge of the states visited and the actions taken by the expert. We purely rely on observations, $o \in O$, provided in a set of expert demonstrations $B_E = \{\tau_1', \tau_2', ..., \tau_N'\}$ containing visual trajectories $\tau_i' = \left(o_0^i, o_1^i, ..., o_T^i\right)$. Each visual trajectory $\tau_i'$ is a sequence of observations obtained by watching the expert act in its *domain*. Here, the discriminator $D$ will be optimized to discern between the expert demonstrations in $B_E$ and 'agent' observations in $B_\pi$, a set of visual trajectories collected by watching the agent act according to $\pi$. We consider each observation to be an RGB image of the environment at the current time-step, hence, containing only partial and highly-entangled information about the true state. We define the expert's and agent's *domains* as distinct Partially Observed Markov Decision Processes (POMDPs), $M_E = (S_E, A_E, O_E, P, p_o, R)$ and $M_A = (S_A, A_A, O_A, P, p_o, R)$, respectively. The main challenge in *observational* imitation is that the states $s \in S_E$, actions $a \in A_E$ and observations $o \in O_E$ in the expert's POMDP do not necessarily match the states $s \in S_A$, actions $a \in A_A$ and observations $o \in O_A$ in the agent's POMDP.

## 3.3 MUTUAL INFORMATION

To overcome the differences between the expert's and agent's POMDPs, our algorithm relies on estimating the mutual information between different sets of variables. Mutual information is a statistical measure that represents the level of dependency between two random variables and quantifies how much information each random variable is *expected* to contain about the other (Kinney & Atwal, 2014). In other words, the mutual information between $X$ and $Z$ measures the difference between the entropy of $X$ and the conditional entropy of $X$ given $Z$:

$$I(X, Z) = H(X) - H(X|Z) = H(Z) - H(Z|X). \tag{3}$$

Belghazi et al. (2018) showed that the mutual information can be effectively estimated through the Mutual Information Neural Estimator (MINE). This consists in lower bounding the mutual information through the Donskher-Varadhan dual representation of the KL divergence between two random variables distributions (Donsker & Varadhan, 1975), by searching for a parameterized function $T_\phi$:

$$I(X, Z) \geq \sup_{\phi \in \Phi} E_{P(X,Z)}[T_\phi(x, z)] - \log \left( E_{P(X),P(Z)} \left[ e^{T_\phi(x,z)} \right] \right). \tag{4}$$

## 4 DISENTANGAIL

*DisentanGAIL* utilizes several algorithmic components to address the problem of *observational* imitation. Particularly, its discriminator, $D$, is regularized by the enforcement of two mutual information constraints between the domain origin and a specific latent representation of the collected observations. This enables the pseudo-rewards, $R_D$, to disregard the domain differences between the

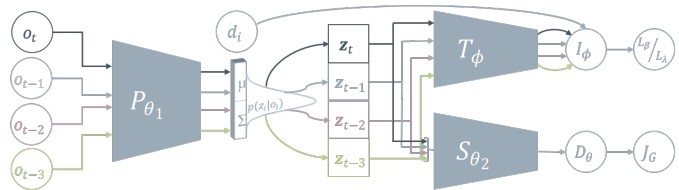

Figure 1: Simplified discriminator optimization structure: for each time-step the four most recent observations $o_{t:t-3}$ are processed independently by the preprocessor $P_{\theta_1}$, outputting the corresponding latent representations $\mathbf{z}_{t:t-3}$. The latent representations are then concatenated and fed jointly into the invariant discriminator $S_{\theta_2}$, and fed individually into the statistics network $T_\phi$, outputting respectively the GAIL objective $J_G$, and the penalty loss $L_\beta$ or $L_\lambda$.

expert's and agent's domain and provide a meaningful learning signal to improve the agent's policy, $\pi$. Together with the visual trajectories in $B_E$ and $B_\pi$, *DisentanGAIL* also exploits sets of prior data collected in the expert's and agent's domains, denoted by $B_{P.E}$ and $B_{P.\pi}$ respectively. Such data is obtained by recording observations from the expert's and agent's domains, while neither expert nor agent is attempting to perform the target task, e.g., while they are acting randomly.

## 4.1 DISCRIMINATOR COMPONENTS

Our algorithm for *observational* imitation utilizes a convolutional neural network discriminator $D_\theta$, optimized for outputting the probability that an input sequence of observations occurred as a consequence of expert behavior. As illustrated in Fig. 1, our discriminator can be divided into two distinct sub-models, namely, the preprocessor $P_{\theta_1}$ and the invariant discriminator $S_{\theta_2}$:

$$D_\theta = S_{\theta_2} \circ P_{\theta_1}. \tag{5}$$

We define the preprocessor as a parameterized multivariate Gaussian distribution with diagonal covariance $P_{\theta_1} = \{\mu_{\theta_1}, \Sigma_{\theta_1}\}$, from which a latent representation is sampled for each input observation $\mathbf{z}_i \sim N(\mu_{\theta_1}(o_i), \Sigma_{\theta_1}(o_i))$. The preprocessor's objective is to project each observation into a latent space containing information about the achievement state of the goal, disregarding the irrelevant information about the inherent differences between the expert's and agent's domains. The Gaussian representation ensures that, for any input, the support over the possible latent representations $\mathbf{z}$ is infinite, moreover, it allows the model to directly reduce the information in any of the independent dimensions of $\mathbf{z}$ by increasing the corresponding variance in $\Sigma_{\theta_1}(o_i)$.

Based on these latent representations, the invariant discriminator $S_{\theta_2}$ is tasked to output the discriminator score for the observed behavior. To classify behavior at any time-step, $S_{\theta_2}$ takes as input the concatenated sequence of the latent representations of the four most recent observations, $\hat{\mathbf{z}}_t = concat(\mathbf{z}_t, \mathbf{z}_{t-1}, \mathbf{z}_{t-2}, \mathbf{z}_{t-3})$. Feeding a concatenation of the latent representations over multiple time-steps serves the purpose of facilitating the recovery of information regarding the true unobserved state of the POMDP from the observations. It also allows the discriminator to reason directly with goal-completion progress throughout different consecutive observations. To understand the necessity of this practice, consider a navigation problem where the agent's task is to reach a target position. Only given access to information about multiple visual observations showing the location of the agent at different time-steps, the discriminator will be able to assess if the agent is approaching the target and retrieve higher-order state information about its motion.

Both the preprocessor and invariant discriminator are trained end-to-end through the reparameterization trick (Kingma & Welling, 2013), to optimize the GAIL objective $J_G$ to discern behavior from the set of expert demonstrations $B_E$ against behavior from the set of recent agent observations $B_\pi$:

$$\arg\max_\theta J_G(\theta, B_E, B_\pi) = \arg\max_\theta \mathbb{E}_{B_E, P_{\theta_1}} \log(S_{\theta_2}(\hat{\mathbf{z}}_i)) + \mathbb{E}_{B_\pi, P_{\theta_1}} \log(1 - S_{\theta_2}(\hat{\mathbf{z}}_i)). \tag{6}$$

## 4.2 MUTUAL INFORMATION CONSTRAINTS

To obtain an invariant latent space from the preprocessor's output, we propose to enforce two different constraints on the mutual information between the observation's latent representations $\mathbf{z_i} \sim P_{\theta_1}(o_i)$ and a corresponding set of domain labels. Each domain label $d_i$ is a binary variable representing whether the associated observation $o_i$ was collected in the expert POMDP, i.e.,

$d_i = 1_{o_i \in B_E \cup B_{P.E}}$. To estimate the mutual information, we make use of the MINE estimator and utilize a statistics network $T_\phi$ optimized to maximize the objective in Eq. 4 between the latent representations and the domain labels for the observations in $B_E$ and $B_\pi$:

$$\arg\max_\phi I_\phi(\mathbf{z}_i, d_i | B_E \cup B_\pi) = \arg\max_\phi \mathbb{E}_{P(d_i), P(\mathbf{z}_i | d_i)}\left[T_\phi(\mathbf{z}_i, d_i)\right] - \log\left(\mathbb{E}_{P(d_i), P(\mathbf{z}_i)}\left[e^{T_\phi(\mathbf{z}_i, d_i)}\right]\right). \quad (7)$$

**Expert demonstrations constraint**. The first mutual information constraint is for the latent representations of the observations from the union of $B_E$ with $B_\pi$. We propose to constraint the estimated mutual information of these latent representations with the corresponding domain labels to be less than 1 bit: $I_\phi(\mathbf{z}_i, d_i | B_E \cup B_\pi) < 1$. We define two kinds of information that the preprocessor $P_{\theta_1}$ can encode into the latent representations to aid the invariant discriminator $S_{\theta_2}$ in discerning transitions $o_{t:t-3}$ from $B_E$ and $B_\pi$: (i) *domain information*, from the visual differences of the two environments (labeled by $d_i$), or (ii) *goal-completion* information, from the expected progress shown in the observations towards achieving the goal demonstrated by the expert in $B_E$ (represented by the variable $c_i$). By constraining the mutual information of the latent representations with the domain labels $d_i$ to be less than 1 bit, we prevent the invariant discriminator to exclusively rely on information inherent to the domain origin to make its classification decision. Therefore, we force it to seek *goal-completion* information about $c_i$ to fully optimize its objective from Eq. 6. We empirically evaluate this constraint against tighter constraints in Appendix D.

**Prior data constraint**. An additional mutual information constraint is for the latent representations of the observations from the union of prior data sets collected independently in both agent and expert domains, $B_{P.E}$ and $B_{P.\pi}$. The observations collected in these sets are expected to come from observing both expert's and agent's domains, while neither expert or agent are attempting to perform the target task. Hence, by assuming that the *goal-completion* levels observed in these two sets approximately match, we can constraint the mutual information of the relative latent representations with the domain labels to be near 0, namely $I_\phi(\mathbf{z}_i, d_i | B_{P.E} \cup B_{P.\pi}) \approx 0$. This mutual information constraint implicitly optimizes for a mapping which makes the distributions of latent representations generated from the observations in the two prior sets of data equivalent. Hence, it allows for the utilization of great amounts of cheaply collected unsupervised data to provide an additional learning signal regarding the information which should be discarded by the preprocessor.

**Comparison with prior efforts.** Stadie et al. (2017) proposed to constraint the mutual information between the encoded observations and the domain labels in $B_E \cup B_\pi$ to be 0. We argue that enforcing such constraint would unnecessarily limit the information in the latent representations and impair learning. This constraint assumes that some observable factor determining $c_i$ is independent of the domain labels $d_i$, otherwise, no information about $c_i$ can be encoded in the latent representations. However, given that in $B_E \cup B_\pi$ we have $o_i \in B_E \Leftrightarrow d_i = 1$, such assumption seldom holds, as it requires the distributions of some *goal-completion* information about $c_i$ present in the observations in $B_E$ and $B_\pi$ to exactly match. However, unlike this work, the algorithm proposed by Stadie et al. (2017) does not attempt to enforce such constraint precisely. Instead, it simply penalizes a measure proportional to the mutual information via a domain confusion loss with a fixed weight coefficient. We further discuss the implications of this practice and provide a toy example where truly enforcing such constraint would prevent any learning in Appendix A.

### 4.3 OFF-POLICY LEARNING

To learn effective behaviour, we combine our regularized *DisentanGAIL* discriminator with the off-policy Soft-Actor Critic (SAC) algorithm by Haarnoja et al. (2018b). To optimize a parameterized agent policy, $\pi_\omega$, SAC maximizes the expected sum of entropy regularized pseudo-rewards:

$$\arg\max_\omega J(\omega) = \arg\max_\omega \mathbb{E}_{p_{\pi_\omega(\tau)}}\left[\sum_{t=0}^{T} R_D(o_t, o_{t-1}, o_{t-2}, o_{t-3}) - \alpha\pi_\omega(a_t | s_t)\right]. \quad (8)$$

## 5 IMPLEMENTATION

### 5.1 ENFORCING THE MUTUAL INFORMATION CONSTRAINTS

We implement two different techniques to enforce the mutual information constraints proposed in Section 4.2, given a set of observations $B$ with the corresponding domain labels. Both techniques

make use of a single hyper-parameter $I_{max}$, which represents the upper limit on the estimated information about the domain labels that we allow the latent representations to retain.

**Adaptive penalty** $L_\beta$. The first technique is having a supplementary loss function for the preprocessor $P_{\theta_1}$, penalizing it proportionally to the estimated mutual information in the latent representations. We use an adaptive parameter $\beta$ to ensure the mutual information is within the desired range:

$$L_\beta(\theta_1, B) = \beta I_\phi(\mathbf{z}_i, d_i | B). \tag{9}$$

We design our updates of $\beta$ to follow a similar pattern to the adaptive penalty coefficient proposed by Schulman et al. (2017), utilizing $I_{max}$ and updating:

- $\beta \leftarrow \beta \times 1.5$, if $I_\phi(\mathbf{z}_i, d_i | B) > I_{max}$   •   $\beta \leftarrow \beta \div 1.5$, if $I_\phi(\mathbf{z}_i, d_i | B) < I_{max} \div 2$.

**Dual penalty** $L_\lambda$. The second technique consists in a different supplementary loss function penalizing the preprocessor $P_{\theta_1}$ proportionally to the violation of the upper limit. In this case, we ensure constraint enforcement through the introduction of a non-negative Lagrange multiplier variable $\lambda$:

$$L_\lambda(\theta_1, B) = \lambda \left( I_\phi \left( \mathbf{z}_i, d_i | B \right) - I_{max} \right) \tag{10}$$

where $\lambda$ is updated to maximize $L_\lambda$, approximating dual gradient descent (Boyd et al., 2004):

$$\lambda \leftarrow \max \left( 0, \lambda + \alpha \left( I_\phi \left( \mathbf{z}_i, d_i | B \right) - I_{max} \right) \right). \tag{11}$$

In practice, the dual penalty enforces precisely the mutual information constraints, but the dual variable $\lambda$ stabilizes in more iterations than the adaptive parameter $\beta$. Our final implementation makes use of the adaptive penalty when enforcing the expert demonstrations constraint with $I_{max} = 0.99$ and the dual penalty when enforcing the prior data constraint with $I_{max} = 0.001$. Hence, our penalized discriminator objective augments the original discriminator objective in Eq. 6 as:

$$\arg\max_\theta J_G(\theta, B_E \cup B_\pi) - L_\beta(\theta_1, B_E \cup B_\pi) - L_\lambda(\theta_1, B_{P.E} \cup B_{P.\pi}). \tag{12}$$

## 5.2 DOMAIN INFORMATION DISGUISING

The discriminator $D_\theta$ is updated to maximize Eq. 12, given a mutual information estimate from a fixed-sized statistics network $T_\phi$. Hence, if the optimization of $T_\phi$ temporarily converges to a sub-optimal local minimum, $D_\theta$ could encode domain information into the latent representations $\mathbf{z}$ without the statistics network detecting it. We refer to this phenomenon as *domain information disguising*, and we utilize two further techniques to prevent this issue.

**Double statistics network**. The first technique takes inspiration from the work of Van Hasselt et al. (2016) and consists of learning independently two statistics neural networks, namely $T_{\phi_1}$ and $T_{\phi_2}$. Thus, the mutual information is estimated through taking the maximum prediction of the independent models over the same set of observations, $\hat{I}_\phi(\mathbf{z}_i, d_i | B) = \max(I_{\phi_1}(\mathbf{z}_i, d_i | B), I_{\phi_2}(\mathbf{z}_i, d_i | B))$. This change makes it impractical for the discriminator to disguise domain information, as the gradient from each of the regularization losses can only contain information about a single statistics network at a time (from the $\max$ operation). Therefore, this gives a statistics network reaching a sub-optimal local minimum the chance to recover, without affecting the mutual information estimates to a great extent. This practice has also the benefit of providing a better prediction of the current mutual information, counteracting the effects of epistemic uncertainty on the optimization.

**Invariant discriminator regularization**. The second technique comprises regularizing the invariant discriminator $S_{\theta_2}$ to be approximately 1-Lipschitz. A great part of the GAN literature (Arjovsky et al., 2017; Gulrajani et al., 2017; Miyato et al., 2018) showed the effectiveness of this practice when regularizing discriminator networks. In our specific problem setting, it has the further benefit of restricting the expressivity of the invariant discriminator, preventing it from capturing domain information not captured by our mutual information estimator. To enforce this, we utilize spectral normalization, a regularization technique proposed by Miyato et al. (2018).

*DisentanGAIL* trains all the models end-to-end, in three main learning steps: (i) *Discriminator learning*, where the discriminator's parameters $\theta$ are updated to maximize Eq. 12; (ii) *Mutual information learning*, where the statistics network's parameters $\phi$ are updated to maximize Eq. 7; (iii) *Agent learning*, where the learner's parameters $\omega$ are updated to maximize Eq. 8 with SAC. We provide further implementation details and a formal summary of the algorithm in Appendix B.

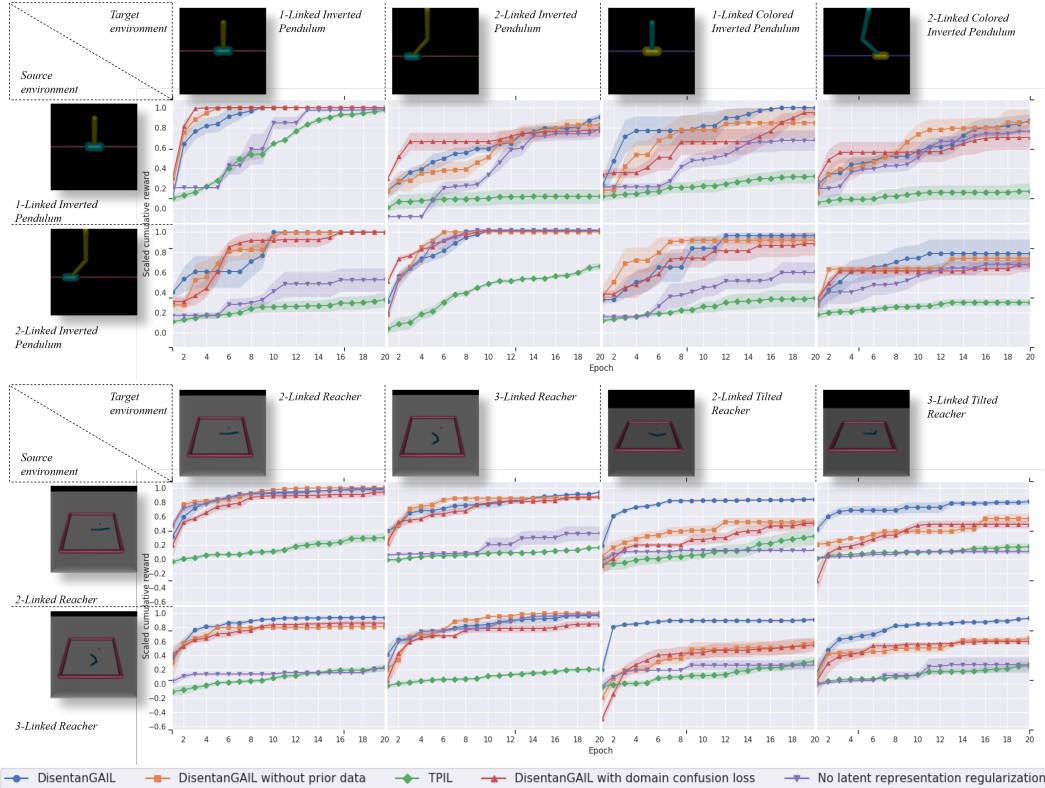

Figure 2: Performance curves for the *Inverted Pendulum* (Top) and *Reacher* (Bottom) realms. *DisentanGAIL* is the only algorithm consistently achieving a performance close to the 'expert' agent's.

Table 1: Results summary for the *Inverted Pendulum* and *Reacher* environment realms

| | Differences between the agent and the expert domains | | | | | | | |
|---|---|---|---|---|---|---|---|---|
| | No differences | | Embodiment | | Appearance | | Embodiment and appearance | |
| Algorithms evaluated: | Reacher | Inverted Pendulum | Reacher | Inverted Pendulum | Reacher | Inverted Pendulum | Reacher | Inverted Pendulum |
| DisentanGAIL | $0.973 \pm 0.074$ | $1.021 \pm 0.023$ | $\mathbf{0.941 \pm 0.045}$ | $\mathbf{0.954 \pm 0.081}$ | $\mathbf{0.885 \pm 0.064}$ | $\mathbf{0.894 \pm 0.231}$ | $\mathbf{0.860 \pm 0.081}$ | $\mathbf{0.918 \pm 0.115}$ |
| DisentanGAIL (No prior) | $\mathbf{1.004 \pm 0.012}$ | $1.015 \pm 0.023$ | $0.847 \pm 0.064$ | $0.914 \pm 0.134$ | $0.586 \pm 0.143$ | $0.794 \pm 0.234$ | $0.578 \pm 0.160$ | $0.887 \pm 0.195$ |
| TPIL | $0.251 \pm 0.111$ | $0.812 \pm 0.162$ | $0.185 \pm 0.079$ | $0.218 \pm 0.191$ | $0.278 \pm 0.217$ | $0.309 \pm 0.122$ | $0.235 \pm 0.154$ | $0.254 \pm 0.199$ |
| TPIL ($\times 5$ experience) | $0.683 \pm 0.158$ | $1.024 \pm 0.025$ | $0.493 \pm 0.195$ | $0.331 \pm 0.279$ | $0.585 \pm 0.256$ | $0.519 \pm 0.281$ | $0.626 \pm 0.282$ | $0.313 \pm 0.266$ |
| DisentanGAIL (DCL) | $0.894 \pm 0.134$ | $\mathbf{1.024 \pm 0.025}$ | $0.867 \pm 0.071$ | $0.889 \pm 0.159$ | $0.550 \pm 0.146$ | $0.826 \pm 0.194$ | $0.523 \pm 0.177$ | $0.786 \pm 0.288$ |
| No regularization | $0.988 \pm 0.042$ | $1.018 \pm 0.038$ | $0.290 \pm 0.187$ | $0.635 \pm 0.230$ | $0.200 \pm 0.176$ | $0.677 \pm 0.178$ | $0.186 \pm 0.136$ | $0.682 \pm 0.182$ |

## 6 EXPERIMENTS

To evaluate our algorithm, we design six different *environment realms*, simulated with Mujoco (Todorov et al., 2012), extending the environments from Brockman et al. (2016): *Inverted Pendulum*, *Reacher*, *Hopper*, *Half-Cheetah*, *7DOF-Pusher* and *7DOF-Striker*. We define an *environment realm* as a set of environments with a shared semantic goal but with significant differences in terms of appearance and agent embodiment. For each of the experiments, we select a *source* environment and a *target* environment within one environment realm. Thus, we train an 'expert' agent and collect a set of visual trajectories in the *source* environment. An 'observer' agent will then use these visual trajectories to perform imitation in the *target* environment, without access to the reward function. In all our experiments, each epoch corresponds to the 'observer' agent collecting 1000 time-steps of experience in the *target* environment. We report at each epoch the mean and standard error over five experiments of the maximum expected cumulative reward recorded so far. We obtain the expected cumulative reward by averaging the performance of the 'observer' agent over five trajectories. We scale the cumulative rewards such that 0 represents the performance from random behavior, and 1 represents the performance obtained by the 'expert' agent. We provide a detailed description of the different environment realms in Appendix C.

**Can *DisentanGAIL* efficiently solve the problem of *observational* imitation with both appearance and embodiment mismatches?** We first evaluate the performance of our algorithm on the

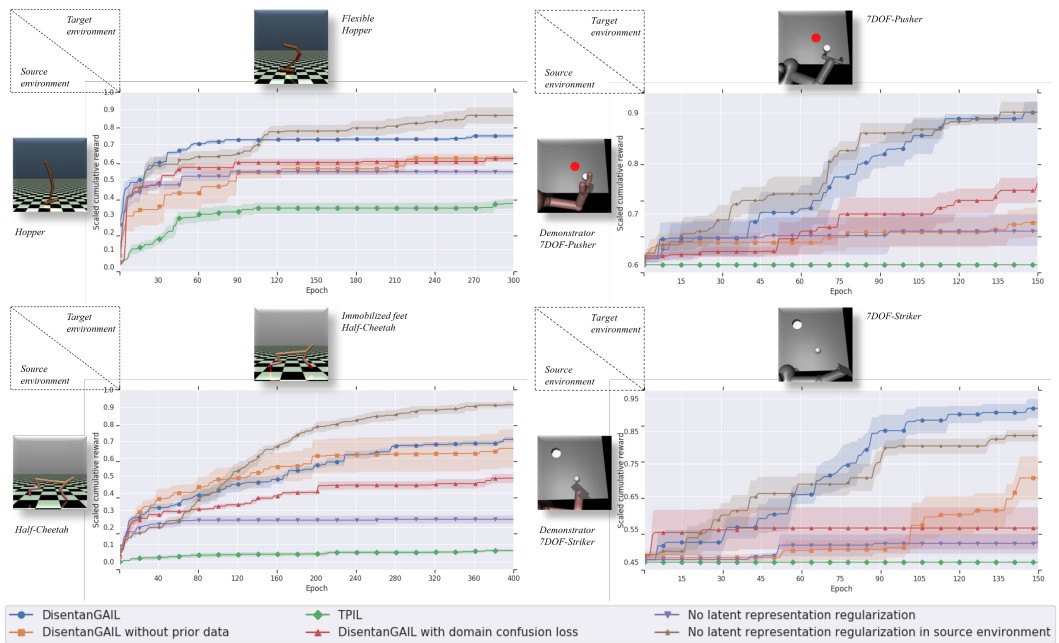

Figure 3: Performance curves for the *Hopper* (Top-left), *Half-Cheetah* (Bottom-left), *7DOF-Pusher* (Top-right) and *7DOF-Striker* (Bottom-right) environment realms.

*Inverted Pendulum* and *Reacher* realms. We test eight different combinations of *source* and *target* environments for each of these realms. We allow the 'observer' agent to train for a maximum of 20 epochs. To evaluate the effectiveness of our proposed constraints and techniques in solving the problem of *observational* imitation, we compare the performance of the following algorithms:

- *DisentanGAIL*: The full proposed algorithm, as described in Section 5.

- *DisentanGAIL without prior data (No prior)*: *DisentanGAIL* without the *prior data* constraint.

- *TPIL*: The original implementation of the algorithm from Stadie et al. (2017).

- *DisentanGAIL with domain confusion loss (DCL)*: Re-implementation of the *domain confusion loss* by Stadie et al. (2017) applied to *DisentanGAIL*, substituting the proposed constraints.

- *No latent representation regularization (No regularization)*: *DisentanGAIL* model without any loss or constraint to prevent encoding domain information in its latent representations.

We present the performance curves in Fig. 2 and a summary of the results in Table 1. Particularly, the full *DisentanGAIL* algorithm outperforms all other algorithms when considering any domain difference and consistently achieves a performance close to the 'expert' agent. In comparison, *TPIL* severely under-performs, even when evaluated given five times the amount of experience. Applying the *domain confusion loss* to *DisentanGAIL* significantly and consistently deteriorates the performance, validating the effectiveness of our proposed constraints. However, this version of *DisentanGAIL* still vastly outperforms the original *TPIL* implementation, underlying the superiority of our proposed model and optimization. *DisentanGAIL with no prior data*, performs well in most experiments, but under-performs when faced with drastic changes in domain appearance, indicating that the utilization of sets of prior data is important when strong visual cues about environment correspondences are missing. We report additional ablation studies for our model in Appendix D.

**Can *DisentanGAIL* scale to more challenging, high dimensional control tasks?** We evaluate our algorithm on the four remaining realms, which consist of substantially harder problems, narrowing the evaluation gap with *agent-centric* imitation algorithms. We refer to the environments in these realms as 'high dimensional' since their state and action spaces are significantly larger than the state and action spaces of the environments explored in prior work making use of the *domain confusion loss* (Stadie et al., 2017; Okumura et al., 2020; Choi et al., 2020). Namely, we explore two locomotion realms, *Hopper* and *Half-Cheetah*, and two manipulation realms, *7DOF-Pusher* and

Table 2: Results summary for the 'high dimensional' environment realms

| | Environment realms | | | |
|---|---|---|---|---|
| *Algorithms evaluated:* | *Hopper* | *Half-Cheetah* | *7DOF-Pusher* | *7DOF-Striker* |
| *DisentanGAIL* | **0.749 ± 0.026** | **0.712 ± 0.036** | **0.901 ± 0.044** | **0.921 ± 0.061** |
| *DisentanGAIL (No prior)* | 0.622 ± 0.051 | 0.660 ± 0.231 | 0.677 ± 0.072 | 0.707 ± 0.150 |
| *TPIL* | 0.362 ± 0.057 | 0.066 ± 0.020 | −0.020 ± 0.068 | 0.081 ± 0.065 |
| *DisentanGAIL (DCL)* | 0.619 ± 0.036 | 0.486 ± 0.058 | 0.747 ± 0.054 | 0.554 ± 0.134 |
| *No regularization* | 0.543 ± 0.039 | 0.247 ± 0.052 | 0.657 ± 0.080 | 0.504 ± 0.069 |
| *No regularization (source)* | *0.866 ± 0.100* | *0.914 ± 0.041* | *0.901 ± 0.044* | *0.837 ± 0.038* |

*7DOF-Striker*. We use *DisentanGAIL* to perform *observational* imitation with the 'source' and 'target' environments differing greatly both in terms of appearance and agent embodiment, as detailed in Appendix C. We compare *DisentanGAIL* with the previously-introduced baselines. To provide an upper bound on the expected performance of *DisentanGAIL*, we additionally evaluate the *No latent representation regularization* baseline with the 'observer' agent learning in the original 'source' environment, i.e., imitating with no domain differences.

We present the performance curves in Fig. 3 and a summary of the results in Table 2. Remarkably, *DisentanGAIL* is able to recover close to the expert's performance in both manipulation realms, with at least the same efficiency as the *No latent representation regularization* baseline learning in the 'source' environment. In the locomotion realms, the performance appears to converge more slowly to a similar but lower value than the expert. We hypothesize this is because the main objectives in the locomotion realms are based on the agents continuously executing a particular stream of actions rather than reaching a target state. Thus, the analyzed shifts in agent's embodiment, even modifying the action-spaces dimensionality, strongly increase the discriminator ambiguity on rewarding the best possible way to solve the tasks. The performance gap of *DisentanGAIL* with the rest of the baselines is considerably greater than in the previous set of experiments. In particular, *TPIL* and the *No latent representation regularization* baseline fail to recover meaningful behavior in any experiment. Similarly, applying the *domain confusion loss* to *DisentanGAIL* degrades considerably the performance across all problems. Additionally, removing the prior data constraint from *DisentanGAIL* also appears to degrade the performance. Yet, *DisentanGAIL with no prior data* still outperforms all other baselines in three environment realms and is able to almost match the full *DisentanGAIL* performance in the *Half-Cheetah* realm. These results highlight the complexity of performing *observational* imitation in high dimensional environments and show the effectiveness of our proposed constraints and optimization.

# 7 CONCLUSION

We proposed *DisentanGAIL* – a novel algorithm to effectively solve the problem of *observational* imitation. Our method makes use of two mutual information constraints for a latent representation inside the discriminator network to encode *goal-completion* information and discard *domain* information about the observations. Unlike prior work, our experiments show *DisentanGAIL*'s effectiveness at dealing with various domain differences, both in terms of environment appearance and agent embodiment, and at scaling to more complex high dimensional tasks. We believe our work might have strong implications for future real-world imitation learning, as it could allow users to teach agents new tasks by simply being observed, leading to natural human-robot interactions. To facilitate future efforts, we share the code for our algorithms and environments: `https://github.com/Aladoro/domain-robust-visual-il`.

ACKNOWLEDGMENTS

Edoardo Cetin would like to acknowledge the support from the Engineering and Physical Sciences Research Council [EP/R513064/1].

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

# A  ALTERNATIVE EXPERT DEMONSTRATIONS CONSTRAINT

Table 3: Example visual trajectories collected in the sample task

| visual trajectories | $\{x_0 y_0, x_1 y_1, x_2 y_2\}$ |
|---|---|
| $B_E$: | $\{10, 11, 11\}$ |
|  | $\{10, 11, 11\}$ |
|  | $\{10, 11, 11\}$ |
|  | $\{10, 11, 11\}$ |
| $B_\pi$: | $\{00, 00, 00\}$ |
|  | $\{00, 01, 00\}$ |
|  | $\{00, 00, 01\}$ |
|  | $\{00, 00, 00\}$ |

Previous work (Stadie et al., 2017) proposed to optimize the GAIL objective described in section 6, subject to constraining the mutual information to be 0:

$$\arg\max_{\theta} J_G(\theta, B_E \cup B_\pi) \; s.t. \; I(\mathbf{z_i}, d_i | B_E \cup B_\pi) = 0.$$

In practice, however, the algorithm proposed by Stadie et al. (2017) enforces such constraint very loosely via a domain confusion loss. This is achieved by introducing a second classifier, $C_\phi$, on top of the preprocessor $P_{\theta_1}$, to predict the domain labels $d_i$ from the latent representations $\mathbf{z}_i$.

$$J_{DCL}(\phi, \theta_1, B_E, B_\pi) = \mathbb{E}_{B_E, P_{\theta_1}} \log(C_\phi(\mathbf{z}_i)) + \mathbb{E}_{B_\pi, P_{\theta_1}} \log(1 - C_\phi(\mathbf{z}_i)).$$

The GAIL objective $J_G$ is then augmented by the domain confusion loss, $J_{DCL}$, where the preprocessor is adversarially trained to minimize the information about the domain labels $d_i$ useful for $C_\phi$. This optimization is then regulated by a fixed weight coefficient $\lambda$:

$$\arg\max_{\theta} \arg\min_{\phi} J_G(\theta, B_E \cup B_\pi) - \lambda J_{DCL}(\phi, \theta_1, B_E, B_\pi).$$

As a consequence, in practice the domain confusion loss acts more as a heuristic to minimize the domain labels information, contained in the single latent representations, rather than attempting to enforce a precise constraint. Below, we provide a toy example where truly enforcing $I(\mathbf{z}_i, d_i) = 0$ would prevent the preprocessor from encoding any useful information about the observations.

Consider a simple task where the objective is for an agent to reach and remain in a target state. In this setting, we let the agent and expert POMDPs differ in their observation spaces. Specifically, we define the observations collected to be composed of two binary variables, $o_i = x_i y_i$. The value of the first variable $x_i$ is 1 for any observation in the expert POMDP, and it is 0 for any observation in the agent POMDP. The value of the second variable $y_i$ is 1 if the visited state is a target state, and it is 0 otherwise. Therefore, for any observation $o_i$ the first binary variable $x_i$ contains *domain* information about $d_i$ which should be discarded (as $x_i = d_i$), while the second binary variable $y_i$ contains useful *goal-completion* information about $c_i$ which should be encoded in $\mathbf{z}_i$.

Consider an instance of this problem with a task horizon of 3 and with four visual trajectories in $B_E$ and $B_\pi$ described in Table 3. In this example, unlike the visual trajectories collected by the agent in $B_\pi$, the expert demonstrations in $B_E$ accomplish the goal of this task as they show successfully reaching and remaining in a target state. Therefore, the distribution of goal states encountered in the observations from $B_E$ and $B_\pi$ are different, and consequently, $y_i$ is not statistically independent from $x_i$. We show this by computing the conditional probabilities:

$$p(x_i = 1 | y_i = 1) = 4/5 \neq p(x_i = 1 | y_i = 0) = 2/7 \Rightarrow p(x_i = 1 | y_i) \neq p(x_i = 1) = 1/2.$$

Thus, the observable goal completion information about $c_i$, present in $y_i$, are not statistically independent from the domain labels $d_i$.

For simplicity, consider a deterministic preprocessor encoding the latent representations as $\mathbf{z}_i = P(o_i)$ (as proposed by Stadie et al. (2017)). Enforcing $I(\mathbf{z}_i, d_i) = 0$ means that the value of $\mathbf{z}_i$ must be independent of $d_i$, and since $d_i = x_i$, we must have $\mathbf{z}_i = P(o_i) = P(xy_i)\forall x = P_y(y_i)$. However, we claim that this also implies that $I(\mathbf{z}_i, c_i) = 0$. We can easily show this by contradiction, assume that $I(\mathbf{z}_i, c_i) > 0$, since $y_i$ is the only observable source of information about $c_i$ then $P_y(y_i = 0) \neq P_y(y_i = 1)$. Therefore, $P_y$ must be invertible or in other words there exists a function such that $P_y^{-1}(P_y(y_i)) = y_i\forall y_i$. However, we then have that $p(d_i|\mathbf{z}_i) = p(d_i|P_y^{-1}(\mathbf{z}_i)) = p(d_i|y_i) = p(x_i|y_i) \neq p(x_i) = p(d_i)$, therefore $\mathbf{z}_i$ is not independent of $d_i$ and we must have $I(\mathbf{z}_i, d_i) \neq 0$.

# B  ALGORITHM DETAILS

## B.1  PRIOR DATA

The prior data sets utilized to enforce the prior data constraint are collected by executing random behavior in both 'source' and 'target' environments. To add diversity to the discriminator's learning signal, we also use samples from both prior sets as additional negative examples for $J_G$. The baselines making use of the domain confusion loss also make use of prior data, analogously to how 'failure' data is used in the original *TPIL* algorithm (Stadie et al., 2017). On the other hand, the baselines *DisentanGAIL with no prior data* and *No latent representation regularization* are evaluated without access to prior data.

## B.2  DISCRIMINATOR

Given an observation $o_i$, to sample the latent representation $\mathbf{z}_i \sim N\left(\mu_{\theta_1}(o_i), \Sigma_{\theta_1}(o_i)\right)$ we flatten the output of the preprocessor and split the resulting $K$-dimensional vector in two halves. We utilize the first half of the variables to obtain the latent representation's mean, while we apply a *Tanh* nonlinearity, exponentiate and scale the second half to obtain the latent representation's covariance:

$$\sigma_{\theta_1}(o_i) = P_{\theta_1}(o_i)_{1:K/2},$$
$$\Sigma_{\theta_1}(o_i) = diag(exp(tanh(P_{\theta_1}(o_i)_{K/2:K}) \div 2)).$$

To obtain a non-stochastic learning signal for the policy, when calculating the pseudo-reward $R_D(o_i, o_{i-1}, o_{i-2}, o_{i-3}) = \log(D_\theta(o_i, o_{i-1}, o_{i-2}, o_{i-3})) - \log(1 - D_\theta(o_i, o_{i-1}, o_{i-2}, o_{i-3}))$ we set the Gaussian noise to zero, equivalently substituting $D_\theta(o_i, o_{i-1}, o_{i-2}, o_{i-3})$ with:

$$D_{\theta_{det}}(o_i, o_{i-1}, o_{i-2}, o_{i-3}) = S_{\theta_2}(concat(\sigma_{\theta_1}(o_i), \sigma_{\theta_1}(o_{i-1}), \sigma_{\theta_1}(o_{i-2}), \sigma_{\theta_1}(o_{i-3}))).$$

## B.3  TRAINING SPECIFICATIONS

The discriminator loss from Eq. 12 and the MINE estimator objective from Eq. 7 are approximated utilizing batches of transitions of observations $b$, sampled uniformly from the corresponding sets of visual trajectories $B$. Specifically, for all optimizations, we set the batch size $|b| = 128$. We utilize a fixed size agent set of visual trajectories $B_\pi$ and evict old transitions when reaching full capacity, thus, effectively acting as a replay buffer (similarly to Kostrikov et al. (2018)). Throughout all the experiments, we utilize the same 2 hidden-layer fully-connected policy and Q-networks with 256 units and *ReLU* nonlinearities. We keep other model architectures structurally consistent, with a fully-convolutional preprocessor $P_{\theta_1}$, a fully-connected invariant discriminator $S_{\theta_2}$ and fully-connected statistics networks $T_{\phi_i}$. We only vary the depth of the models and the number of filters/units in each layer depending on the environment realm. To avoid having a *biased* pseudo-reward which could provide a learning signal to the agent even without any meaningful discriminator (Kostrikov et al., 2018), we modify the environments by removing terminal states. Thus, each collected visual trajectory has a fixed length, equal to the task-horizon $|\tau|$. We provide the utilized environment-specific hyper-parameters in Table 4, where we specify the buffer sizes in terms of total/maximum number of observations.

We train each model through the Adam optimizer (Kingma & Ba, 2014) with a unique learning rate $\alpha = 0.001$ and momentum parameters $\beta_1 = 0.9$, $\beta_2 = 0.999$. We alternate the collection of a single episode using the current policy $\pi_\omega$, with repeating (i) *Discriminator learning* and (ii)

Table 4: Environment-realm specific hyper-parameters

| Realm | $P_{\theta_1}$ | $S_{\theta_2}$ | $T_\phi$ | $|B_E|$ | $|B_{P.x}|$ | $|B_\pi|$ | $|\tau|$ |
|---|---|---|---|---|---|---|---|
| *Inverted Pendulum* | $2 \times \{16 \times (3,3)\text{-}conv, Tanh, (2,2)\text{-}MaxPool\}$ 
 $\{1 \times (3,3)\text{-}conv, (2,2)\text{-}MaxPool\}$ | $2 \times \{32\text{-}FC, ReLU\}$ 
 $\{1\text{-}FC, Sigmoid\}$ | $2 \times \{32\text{-}FC, Tanh\}$ 
 $\{1\text{-}FC\}$ | 10000 | 10000 | 10000 | 50 |
| *Reacher* | $2 \times \{16 \times (3,3)\text{-}conv, Tanh, (2,2)\text{-}MaxPool\}$ 
 $\{1 \times (3,3)\text{-}conv, (2,2)\text{-}MaxPool\}$ | $2 \times \{32\text{-}FC, ReLU\}$ 
 $\{1\text{-}FC, Sigmoid\}$ | $2 \times \{32\text{-}FC, Tanh\}$ 
 $\{1\text{-}FC\}$ | 10000 | 10000 | 10000 | 50 |
| *Hopper* | $\{16 \times (3,3)\text{-}conv, Tanh, (2,2)\text{-}MaxPool\}$ 
 $\{24 \times (3,3)\text{-}conv, Tanh, (2,2)\text{-}MaxPool\}$ 
 $\{32 \times (3,3)\text{-}conv, Tanh, (2,2)\text{-}MaxPool\}$ 
 $\{48 \times (3,3)\text{-}conv, (2,2)\text{-}MaxPool\}$ | $2 \times \{100\text{-}FC, ReLU\}$ 
 $\{1\text{-}FC, Sigmoid\}$ | $2 \times \{128\text{-}FC, Tanh\}$ 
 $\{1\text{-}FC\}$ | 20000 | 20000 | 100000 | 200 |
| *Half-Cheetah* | $\{16 \times (3,3)\text{-}conv, Tanh, (2,2)\text{-}MaxPool\}$ 
 $\{24 \times (3,3)\text{-}conv, Tanh, (2,2)\text{-}MaxPool\}$ 
 $\{32 \times (3,3)\text{-}conv, Tanh, (2,2)\text{-}MaxPool\}$ 
 $\{48 \times (3,3)\text{-}conv, (2,2)\text{-}MaxPool\}$ | $2 \times \{100\text{-}FC, ReLU\}$ 
 $\{1\text{-}FC, Sigmoid\}$ | $2 \times \{128\text{-}FC, Tanh\}$ 
 $\{1\text{-}FC\}$ | 20000 | 20000 | 100000 | 200 |
| *7DOF-Pusher* | $\{24 \times (3,3)\text{-}conv, Tanh, (2,2)\text{-}MaxPool\}$ 
 $\{32 \times (3,3)\text{-}conv, Tanh, (2,2)\text{-}MaxPool\}$ 
 $\{40 \times (3,3)\text{-}conv, Tanh, (2,2)\text{-}MaxPool\}$ 
 $\{64 \times (3,3)\text{-}conv, (2,2)\text{-}MaxPool\}$ | $2 \times \{100\text{-}FC, ReLU\}$ 
 $\{1\text{-}FC, Sigmoid\}$ | $2 \times \{128\text{-}FC, Tanh\}$ 
 $\{1\text{-}FC\}$ | 10000 | 10000 | 100000 | 200 |
| *7DOF-Striker* | $\{24 \times (3,3)\text{-}conv, Tanh, (2,2)\text{-}MaxPool\}$ 
 $\{32 \times (3,3)\text{-}conv, Tanh, (2,2)\text{-}MaxPool\}$ 
 $\{40 \times (3,3)\text{-}conv, Tanh, (2,2)\text{-}MaxPool\}$ 
 $\{64 \times (3,3)\text{-}conv, (2,2)\text{-}MaxPool\}$ | $2 \times \{100\text{-}FC, ReLU\}$ 
 $\{1\text{-}FC, Sigmoid\}$ | $2 \times \{128\text{-}FC, Tanh\}$ 
 $\{1\text{-}FC\}$ | 10000 | 10000 | 100000 | 200 |

*Mutual information learning* for as many iterations as the number of time-steps collected. Then, by averaging the mutual information estimates accumulated from performing (ii), we update the coefficients $\beta$ and $\lambda$. Finally, we also perform (iii) *Agent learning* for as many iterations as the number of time-steps collected. We utilize the same agent set of visual trajectories $B_\pi$ in all different learning steps. A formal summary of *DisentanGAIL* is reported below in Algorithm 1.

---

**Algorithm 1** *DisentanGAIL*

---

1: **Input:** expert demonstrations $B_E$, prior expert and agent observations $B_{P.E}, B_{P.\pi}$
2: Initialize $B_\pi \leftarrow \emptyset$
3: **for** $i = 1, 2, ...,$ **do**
4:      Sample $\tau = (o_t, a_t, s_t)_{t=1}^T$ with $\pi_\omega$
5:      $B_\pi \leftarrow B_\pi \cup \tau$
6:      **for** $j = 1, 2, ..., |\tau|$ **do**
7:          Sample $b_E, b_\pi, b_{P.E}, b_{P.\pi} \in B_E, B_\pi, B_{P.E}, B_{P.\pi}$      ▷ *Discriminator learning*
8:          $L_D = -J_G(\theta, b_E, b_\pi) + L_\beta(\theta_1, b_E \cup b_\pi) + L_\lambda(\theta_1, b_{P.E} \cup b_{P.\pi})$
9:          Update $\theta$ with Adam
10:          **for** $n = 1, 2$ **do**      ▷ *Mutual information learning*
11:             Sample $b_E, b_\pi, b_{P.E}, b_{P.\pi} \in B_E, B_\pi, B_{P.E}, B_{P.\pi}$
12:             $I_n = I_{\phi_n}(\mathbf{z}_i, d_i | b_E \cup b_\pi)$
13:             $I_n^P = I_{\phi_n}(\mathbf{z}_i, d_i | b_{P.E} \cup b_{P.\pi})$
14:             $L_I = -I_n - I_n^P$
15:             Update $\phi_n$ with Adam
16:          $I_j^{est} = \max(I_1, I_2)$
17:          $I_j^{P.est} = \max(I_1^P, I_2^P)$
18:      Update $\beta$ using $\frac{1}{|\tau|} \sum_{j=1}^{|\tau|} I_j^{est}$
19:      Update $\lambda$ using $\frac{1}{|\tau|} \sum_{j=1}^{|\tau|} I_j^{P.est}$
20:      Update $\pi_\omega$ with SAC, sampling from $B_\pi$ for $|\tau|$ steps      ▷ *Agent learning*

---

## C   ENVIRONMENTS DESCRIPTION

We evaluate the algorithms on six different environment realms designed to test the proposed methods for a diverse range of task difficulties and domain difference, extending the environments in OpenAI Gym (Brockman et al., 2016):

Table 5: Description of the implemented environment realms

| Realm | Environment | Characteristics | $dim(A)$ | $dim(O)$ | Semantic goal |
|---|---|---|---|---|---|
| Inverted Pendulum | 1-Linked Inverted Pendulum | 1 link, standard coloring | 1 | $32 \times 32 \times 3$ | Keeping the links |
| | 2-Linked Inverted Pendulum | 2 links, standard coloring | 1 | | vertically balanced |
| | 1-Linked Colored Inverted Pendulum | 1 link, alternative coloring | 1 | | above the moving cart. |
| | 2-Linked Colored Inverted Pendulum | 2 link, alternative coloring | 1 | | |
| Reacher | 2-Linked Reacher | 2 links, standard camera | 2 | $48 \times 48 \times 3$ | Approaching and fixating |
| | 3-Linked Reacher | 3 links, standard camera | 3 | | on a target goal with the |
| | 2-Linked Tilted Reacher | 2 links, camera tilted 14.1 degrees | 2 | | end effector. |
| | 3-Linked Tilted Reacher | 3 links, camera tilted 14.1 degrees | 3 | | |
| Hopper | Hopper | all movable joints, standard coloring | 3 | $64 \times 64 \times 3$ | Hopping forward without |
| | Flexible Hopper | additional thigh joint, alternative thigh coloring | 4 | | falling. |
| Half-Cheetah | Half-Cheetah | all movable joints, standard coloring | 6 | $64 \times 64 \times 3$ | Maximizing forward |
| | Immobilized feet Half-Cheetah | fixed feet joints, alternative feet coloring | 4 | | distance covered. |
| 7DOF-Pusher | 7DOF-Pusher | standard link sizes, standard coloring, 3-linked hand | 7 | $48 \times 48 \times 3$ | Grabbing and pushing an |
| | Demonstrator 7DOF-Pusher | alternative link sizes, alternative coloring, 4-linked hand | 7 | | item to a target goal. |
| 7DOF-Striker | 7DOF-Striker | standard link sizes, standard coloring | 7 | $48 \times 48 \times 3$ | Striking a ball into a |
| | Demonstrator 7DOF-Striker | alternative link sizes, alternative coloring | 7 | | target goal. |

- *Inverted Pendulum*: This environment realm consists of four variations of the original balancing task. The variations explore changing the color of the agent and adding a second link to be balanced on top of the moving cart.

- *Reacher*: This environment realm consists of four variations of the original 2-D reaching task. The variations explore augmenting the number of joints and changing the observer's camera recording angle.

- *Hopper*: This environment realm consists of two environments, including the original Hopper environment and an alternative version, in which the agent has an additional joint splitting its thigh link in two, with the new link also appearing in a different color.

- *Half-Cheetah*: This environment realm consists of two environments, including the original Half-Cheetah environment and an alternative version, in which the agent has immobilized feet joints and the connected links appearing in a different color.

- *7DOF-Pusher*/*7DOF-Striker*: Each of these environment realms consists of two environments, including the original Pusher/Striker environments and an alternative version, in which the agent's model is modified in its appearance and structural configuration, to make it resemble a very simplified human operator.

To collect observations, we render the environments with Mujoco and down-scale the renderings to different dimensions with the purpose of having efficient representations but still preserving the relevant details about the observations. We provide further environment-specific descriptive details in Table 5.

## D SUPPLEMENTARY RESULTS

### D.1 LATENT REPRESENTATIONS COUPLING

After performing the reported experiments, we utilize the learnt models to understand what features are encoded into our constrained latent representations $\mathbf{z}_i$. Particularly, we use the output of the trained preprocessors to map observations between the 'expert' agent's and the 'observer' agent's sets of visual trajectories. We achieve this by taking four different observations from $B_\pi$ and computing their latent representations. Then, we match these observations with the four observations in $B_E$ having the closest latent representations, computed by taking the relative $L1$-distances. We repeat this process, matching four observations in $B_{P,\pi}$ with four observations in $B_{P,E}$. We show the produced couplings for six different experiments with *DisentanGAIL* in Fig. 4. From the results, it can be inferred that the mutual information constraints successfully guide the preprocessor to encode features which are agnostic to the agent's embodiment and the environment's appearance, yet preserving information about the *goal-completion* levels displayed the observations. For the *Inverted Pendulum* realm, the preprocessor appears to be encoding information about the relative

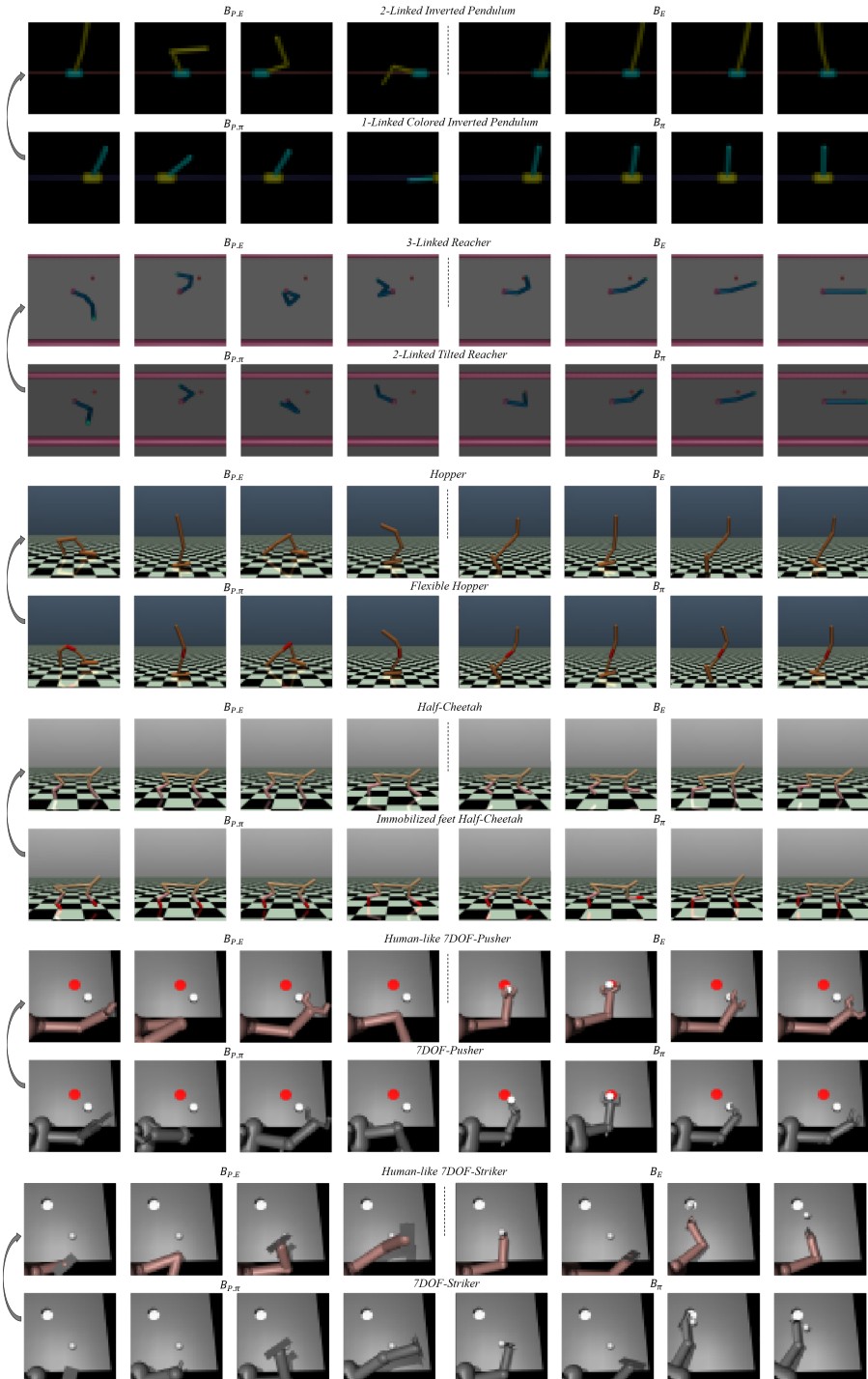

Figure 4: Couplings produced by matching the observations collected in the 'observer' agent's environment with the observations collected in the 'expert' agent's environment, to minimize the $L1$-distance between the latent representations **produced by *DisentanGAIL***. On the right we show the results between the agent set of visual trajectories $B_\pi$ and the set of expert demonstrations $B_E$, and on the left we show the results between the prior set of agent observations $B_{P,\pi}$ and the prior set of expert observations $B_{P,E}$. From top to bottom, we show the results in the *Inverted Pendulum*, *Reacher*, *Hopper*, *Half-Cheetah*, *7DOF-Pusher* and *7DOF-Striker* environment realms. We produce the couplings in six sample *observational* imitation problems considering domain differences in terms of both environment appearance and agent embodiment.

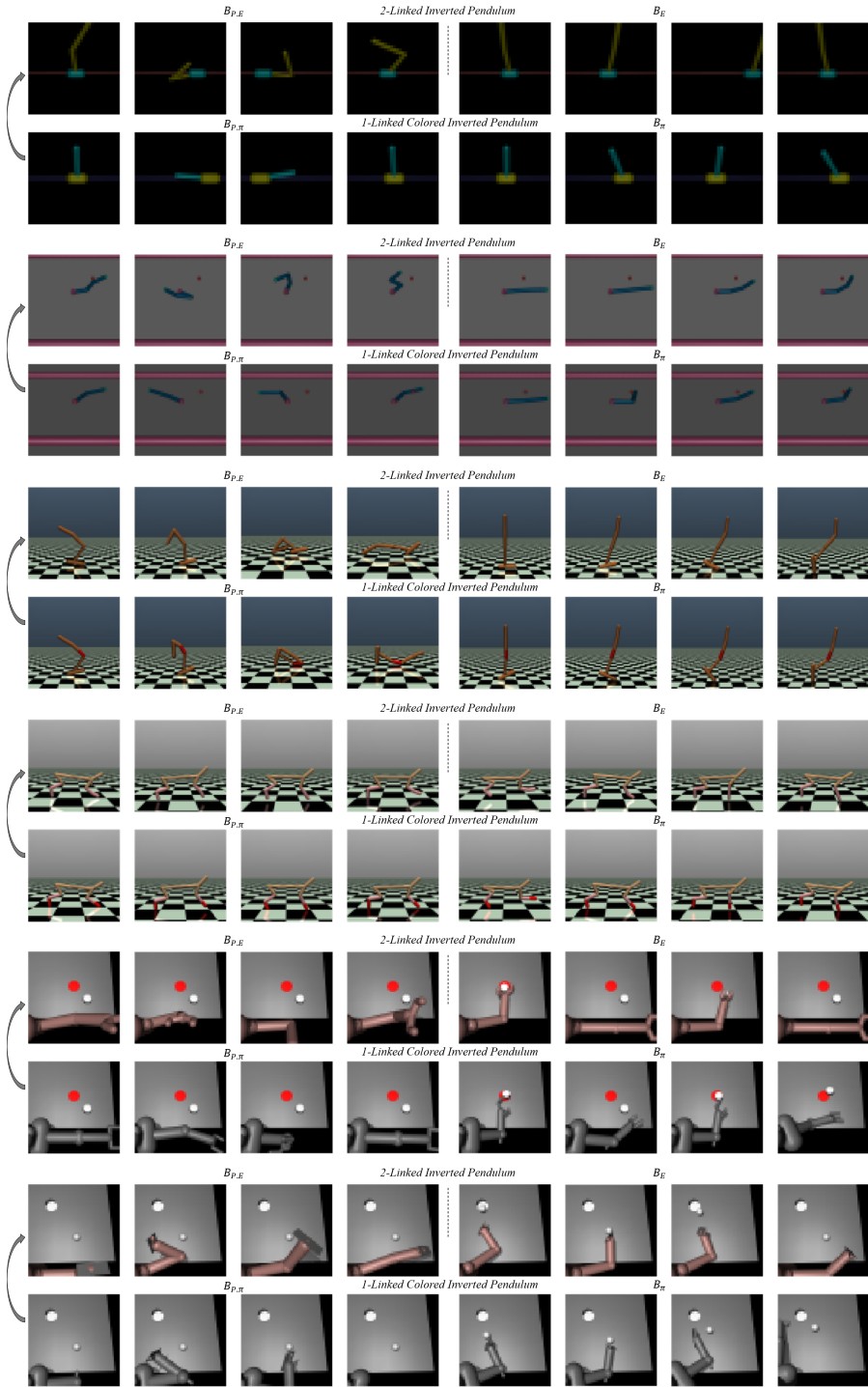

Figure 5: Couplings produced by matching the observations collected in the 'observer' agent's environment with the observations collected in the 'expert' agent's environment, to minimize the $L1$-distance between the latent representations **produced by *DisentanGAIL with domain confusion loss***. This visualization is analogous to Fig. 4

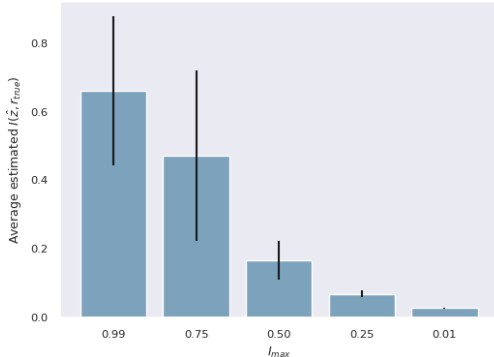

Figure 6: Average mutual information between the latent representations $\hat{\mathbf{z}}$ and the default rewards $r_{true}$ throughout performing *observational* imitation, for differents value of the expert demonstration hyper-parameter $I_{max}$. We evaluate this measure for the experiments in the *Reacher* and *Inverted Pendulum environment realms* considering domain differences in both appearance and embodiment.

angle between the tip of the links and the moving cart, together with some information about the cart's position. For the *Reacher* realm, the preprocessor appears to be encoding information about the position of the tip of the reacher, ignoring the orientation of the individual links. For the *Hopper* and *Half-Cheetah* realms, the preprocessor appears to be encoding the orientation of the joints movable in both domains, together with the current height and position of the agent with respect to the floor tiles, allowing the discriminator to assess whether the agent is hopping/advancing. For the *7DOF-Pusher* and *7DOF-Striker* realms, the preprocessor appears to be encoding the location of the item/ball, together with the location and orientation of the agent's hand actuator.

We also show the produced couplings for six different experiments with *DisentanGAIL with domain confusion loss* in Fig. 5. From the results, it can be inferred that the features encoded by the domain confusion loss are similar to the ones encoded by the mutual information constraints, yet not as consistently interpretable. This is especially evident in the more challenging high dimensional environments. Particularly, for the *Hopper* and *Half-Cheetah* realms, the preprocessor does not appear to be always encoding the agent's relative position to the floor tiles, but rather focusing either on the angle of particular joints or the agent's overall appearance. Additionally, for the *7DOF-Pusher* and *7DOF-Striker* realms, the preprocessor does not appear to be consistently encoding the location of the item/ball, but rather focusing on some less interpretable feature about the agent's appearance.

### D.2 Expert demonstrations constraint

We evaluate the effects of enforcing tighter expert demonstration constraints on *DisentanGAIL* in the low-dimensional environments. This is achieved by running our algorithm with lower values for the hyper-parameter regulating the upper-limit on the estimated mutual information, $I_{max}$, in the adaptive penalty loss $L_{\beta}$ (described in Section 5.1).

First, we analyze the effects that tighter constraints have on the amount of *goal-completion* information encoded in our latent representations $\hat{\mathbf{z}}$. Particularly, for different values of $I_{max}$, we utilize MINE to estimate the mutual information between $\hat{\mathbf{z}}$ and default environment rewards $r_{true}$: $I(\hat{\mathbf{z}}, r_{true})$. We argue that this measure is a good heuristic about the *goal-completion* information contained in $\hat{\mathbf{z}}$ since $r_{true}$ can be effectively used to recover a policy to solve the task in all environments. Specifically, in Fig. 6, we show the average and standard deviation of the mutual information collected throughout different experiments considering domain differences in both appearance and embodiment. This data shows that there is a positive correlation between $I_{max}$ and $I(\hat{\mathbf{z}}, r_{true})$, indicating that tighter mutual information constraints are detrimental. This is explained since, especially at the beginning of training, $c_i$ and $d_i$ are highly dependent. Hence, looser constraint allow greater amounts of information about $c_i$ to be encoded within $\hat{\mathbf{z}}$. These findings are also consistent with our arguments from Appendix A.

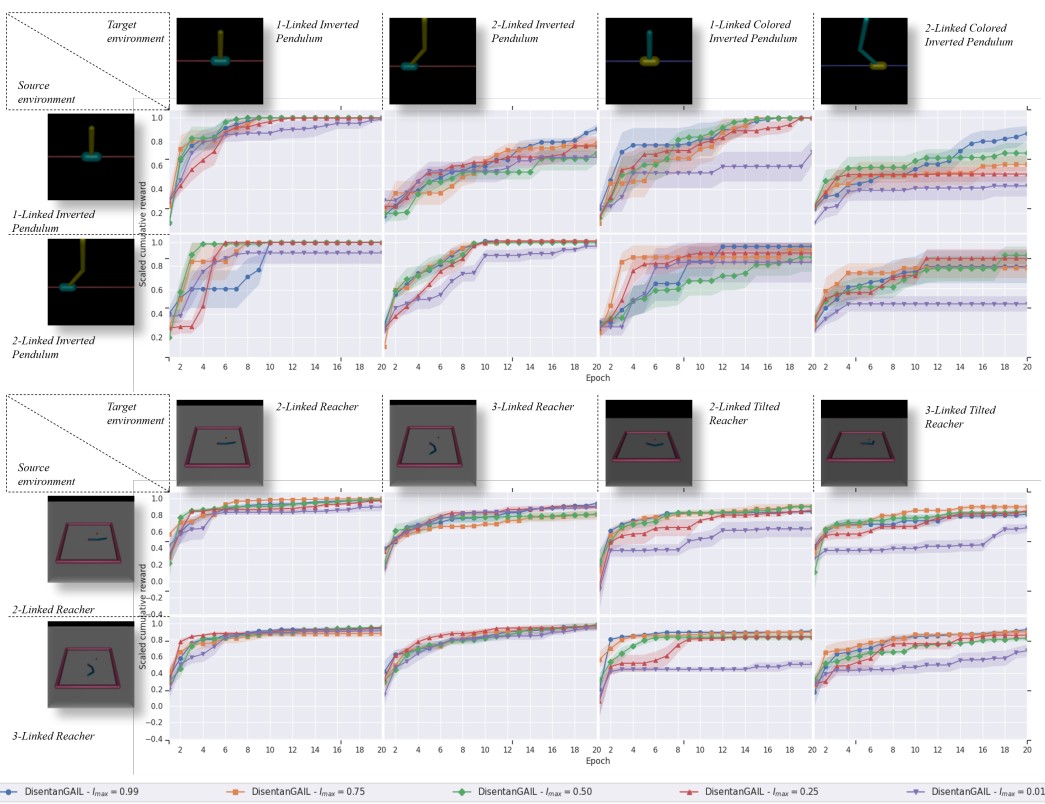

Figure 7: Performance curves from enforcing tighter expert demonstration constraints in the *Inverted Pendulum* (Top) and *Reacher* (Bottom) environment realms.

Table 6: Results summary for *DisentanGAIL* with tighter expert demonstration constraints

| | Differences between the agent and the expert domains | | | | | | | |
| | No differences | | Embodiment | | Appearance | | Embodiment and appearance | |
| *Algorithms evaluated:* | *Reacher* | *Inverted Pendulum* | *Reacher* | *Inverted Pendulum* | *Reacher* | *Inverted Pendulum* | *Reacher* | *Inverted Pendulum* |
|---|---|---|---|---|---|---|---|---|
| *DisentanGAIL–$I_{max}$* = 0.99 | 0.973 ± 0.074 | **1.021 ± 0.023** | **0.941 ± 0.045** | **0.954 ± 0.081** | 0.885 ± 0.064 | 0.894 ± 0.231 | 0.860 ± 0.081 | **0.918 ± 0.115** |
| *DisentanGAIL–$I_{max}$* = 0.75 | 0.983 ± 0.067 | 1.019 ± 0.021 | 0.841 ± 0.113 | 0.892 ± 0.131 | **0.903 ± 0.064** | 0.887 ± 0.180 | **0.897 ± 0.056** | 0.772 ± 0.200 |
| *DisentanGAIL–$I_{max}$* = 0.50 | **0.987 ± 0.045** | 1.013 ± 0.016 | 0.885 ± 0.104 | 0.853 ± 0.208 | 0.861 ± 0.077 | **0.942 ± 0.131** | 0.837 ± 0.071 | 0.790 ± 0.234 |
| *DisentanGAIL–$I_{max}$* = 0.25 | 0.975 ± 0.028 | 1.020 ± 0.025 | 0.927 ± 0.052 | 0.882 ± 0.173 | 0.861 ± 0.088 | 0.930 ± 0.156 | 0.848 ± 0.055 | 0.720 ± 0.251 |
| *DisentanGAIL–$I_{max}$* = 0.01 | 0.921 ± 0.094 | 0.992 ± 0.041 | 0.905 ± 0.057 | 0.790 ± 0.201 | 0.652 ± 0.188 | 0.589 ± 0.224 | 0.576 ± 0.141 | 0.630 ± 0.345 |

Table 7: Results summary for *DisentanGAIL* with missing *domain information disguising* prevention techniques

| | Differences between the agent and the expert domains | | | | | | | |
| | No differences | | Embodiment | | Appearance | | Embodiment and appearance | |
| *Algorithms evaluated:* | *Reacher* | *Inverted Pendulum* | *Reacher* | *Inverted Pendulum* | *Reacher* | *Inverted Pendulum* | *Reacher* | *Inverted Pendulum* |
|---|---|---|---|---|---|---|---|---|
| *DisentanGAIL* | 0.973 ± 0.074 | **1.021 ± 0.023** | **0.941 ± 0.045** | 0.954 ± 0.081 | **0.885 ± 0.064** | **0.894 ± 0.231** | 0.860 ± 0.081 | **0.918 ± 0.115** |
| *DisentanGAIL (No SN)* | 0.957 ± 0.084 | 1.018 ± 0.024 | 0.844 ± 0.105 | 0.937 ± 0.094 | 0.853 ± 0.092 | 0.871 ± 0.153 | **0.861 ± 0.049** | 0.864 ± 0.184 |
| *DisentanGAIL (No 2St)* | **0.982 ± 0.075** | 1.020 ± 0.023 | 0.907 ± 0.109 | 0.900 ± 0.156 | 0.857 ± 0.127 | 0.799 ± 0.199 | 0.858 ± 0.044 | 0.789 ± 0.182 |
| *DisentanGAIL (No Prev)* | 0.949 ± 0.105 | 1.002 ± 0.022 | 0.866 ± 0.056 | **0.969 ± 0.085** | 0.768 ± 0.115 | 0.778 ± 0.235 | 0.763 ± 0.100 | 0.750 ± 0.225 |

Second, we analyze directly the effects that tighter constraints have on the performance of *DisentanGAIL*. The performance curves for different values of $I_{max}$ are shown in Fig. 7 and a summary of the results is given in Table 6. Overall, *DisentanGAIL* appears to be quite robust to all settings tested, excluding the extreme $I_{max} = 0.01$. In general, however, a lower mutual information upper-limit appears to have a negative effect on the performance in most experiments, especially when the 'expert' agent's embodiment and the 'observer' agent's embodiment differ. This is likely because a tighter constraint does not permit the discriminator to utilize enough information about single observations, thus, providing a less informative learning signal to finetune the agent's behavior. The effects of varying $I_{max}$ appear to be less accentuated in the experiments performed in the *Reacher* realm. This is likely because the exploratory policy in this environment covers a greater range of states than in the *Inverted Pendulum* realm, with a more diverse range of *goal-completion* levels. Thus, encoding features carrying *goal-completion* information necessitates to carry less information about the domain labels.

### D.3  DOMAIN INFORMATION DISGUISING

We also perform an ablation study to understand the effects of the techniques proposed to counteract *domain information disguising*, described in Section 5.2. We compare the proposed *DisentanGAIL* algorithm with alternative versions: (i) *with no spectral normalization* (*No SN*) (ii) *with no double statistics network* (*No 2St*) (iii) *with no domain information disguising prevention* (*No Prev* – making use of neither spectral normalization or double statistics network). The performance curves are shown in Fig. 8 and a summary of the results is given in Table 7. Overall, both techniques contribute positively to the final performance. Particularly, the double statistics network appears to have a slightly greater positive effect. This is especially evident in the *Inverted Pendulum* realm. Additionally, removing spectral normalization from the invariant discriminator's layers makes the agent initially learn slightly faster, indicating that there might be a trade-off between convergence speed and training stability.

### D.4  ROBUSTNESS TO BACKGROUND DIFFERENCES

We examine whether *DisentanGAIL*'s performance is affected by larger visual domain differences, solely concerning the background appearance. Particularly, most of the tested domain differences in our environment realms involved changing the appearance and morphology of the agents themselves. Hence, we test *DisentanGAIL* on two alternative target environments in the *Hopper* and *7DOF-Pusher* environment realms, with very distinct backgrounds from the relative source environments. Particularly, the target environment of the *Hopper* realm has a much darker floor, where the tiles are difficult to discern. Additionally, the target environment in the *7DOF-Pusher* realm includes a green table and a white floor, both of which differ greatly in appearance to the grey table and black floor of the source environment. We compare the performance of *DisentanGAIL* performing imitation in these two alternative target environments with the performance in the original target environments to evaluate its robustness.

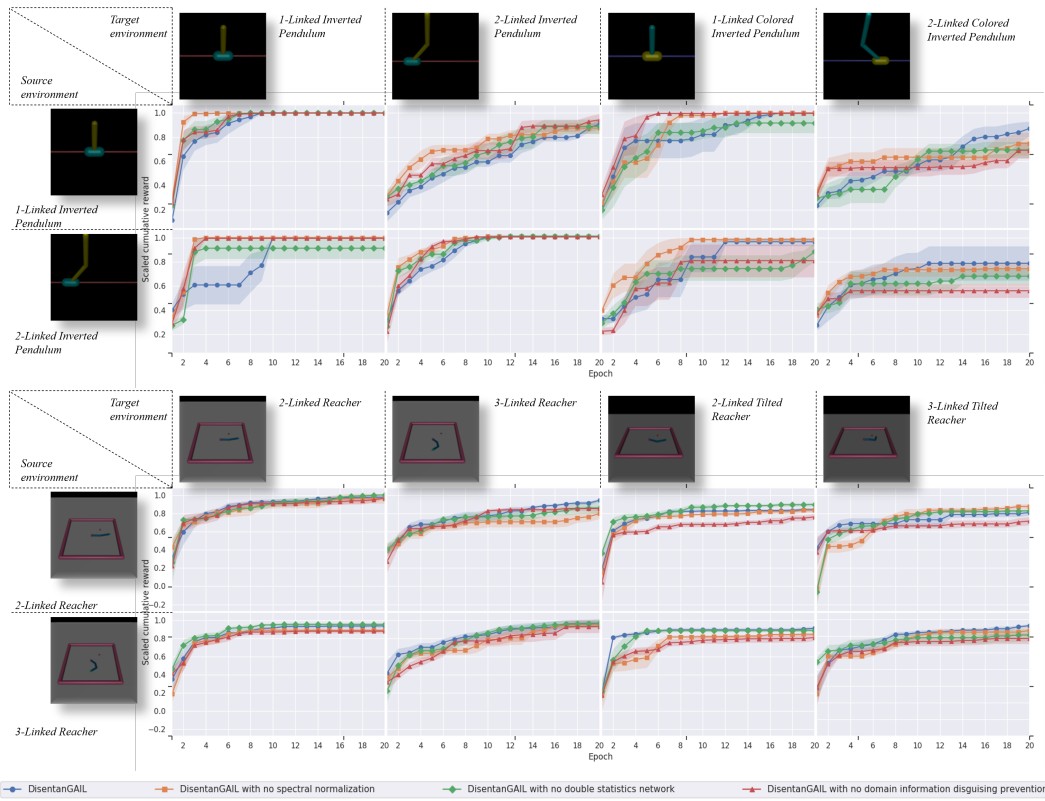

Figure 8: Performance curves for the *domain information disguising* ablation performed in the *Inverted Pendulum* (Top) and *Reacher* (Bottom) environment realms.

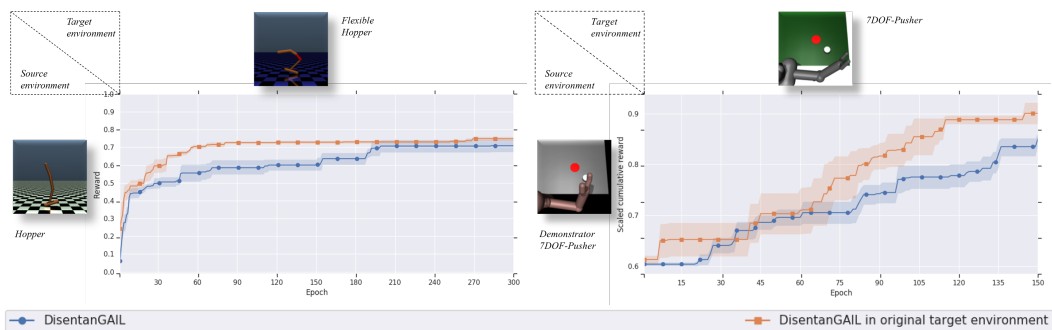

Figure 9: Performance curves for the *Hopper* (Left) and *7DOF-Pusher* (Right) environment realms in the alternative target environments.

Table 8: Results summary for the experiments considering further background domain differences in the 'target' environments

| | Environment realms | |
|---|---|---|
| *Algorithms evaluated:* | *Hopper* | *7DOF-Pusher* |
| *DisentanGAIL* | $0.709 \pm 0.078$ | $0.835 \pm 0.039$ |
| *DisentanGAIL (original target)* | *$0.749 \pm 0.026$* | *$0.901 \pm 0.044$* |

We present the performance curves in Fig. 9 and a summary of the results in Table 8. The obtained results show that background domain differences have a limited effect on *DisentanGAIL*'s final performance. However, they have a more prominent effect on *DisentanGAIL*'s efficiency, making it converge in an increased number of epochs. This is particularly noticeable in the Hopper realm's results. We hypothesize this is because in the new target environments there is a greater amount of domain information that requires to be 'disentangled' from the useful goal-completion information. For example, in the locomotion realms, the pre-processor encodes goal-completion information about the relative position of the agent with respect to the floor tiles in the two environments (as empirically suggested in Section D.1). In the new target environment of the Hopper realm, this information needs to be also disentangled from domain information regarding the tiles' appearance.

