# OpenReview forum: "Domain-Robust Visual Imitation Learning with Mutual Information Constraints"
_ICLR.cc/2021/Conference — ICLR 2021 Poster_

### Official Review · AnonReviewer3 · 2020-10-26
**Strong paper, needs some additional comparisons**

**Rating:** 7
**Confidence:** 4

**Review:**

## Summary
The paper proposes a novel approach for third-person visual imitation learning from observations, ie for imitating a different agent in a potentially different environment purely from visual demonstrations. The main difference over prior work (eg Stadie et al, 2017) that used domain confusion objectives is the introduction of new regularization objectives on the learned representation of the visual scene. Empirically, the paper shows that the proposed regularizations can improve performance across a wide range of third-person visual imitation tasks, transferring between simulated agents with different appearances and/or morphologies.


## Strengths
- the paper is very clearly written and easy to follow, necessary background is adequately covered, all components of the model are explained in detail
- all novel design choices are ablated in the experimental section
- experiments are conducted across multiple environments and source/target variations, both appearance and morphology -- empirical performance improvements are demonstrated
- experiments include higher-dimensional, robotic manipulation environments, more complex than those tested in prior work


## Weaknesses
- **not fully fair comparison to baselines**: since the main novelty lies in the introduction of novel regularization objectives, the "DisentanGAIL w/ domain confusion loss" is the main comparison method since the only difference to the proposed method is the representation regularization function. However, this baseline does not have access to the additional collected prior datasets used for the introduced "prior regularization" loss. Looking at Figure 2, it seems that the performance of DisentanGAIL w/o Prior Data and DisentanGAIL w/ domain confusion loss are near-equivalent which suggests that the access to the additional prior data might be the main factor that contributes to DisentanGAIL's superior performance --> an additional baseline "DisentanGAIL w/ domain confusion loss & prior data" is needed, which additionally trains the domain confusion objective on the prior data collected for the DisentanGAIL prior regularization objective, to allow for fair comparison of both regularization approaches with access to the same data
- **missing baseline results on harder tasks**: on the harder tasks shown in Fig 3 there is no evaluation of the baseline methods, which makes it hard to judge how hard these tasks actually are for prior third-person visual imitation approaches


## Questions
- from my understanding the issue of prior work that constrains the domain-related MI to 0 (described in Appendix, section A) appears for *any* two domains (since, particularly in the beginning of training, there will always be differences in the goal-reaching distributions of expert and policy data). However, it seems that DisentanGAIL w/ domain confusion loss, which applies the MI=0 loss from Stadie et al. 2017, works well on many of the tested domains. How does that fit to the explanation made in Section A + the quant results in Fig 5?
- what differences result in the substantial performance difference between TPIL and DisentanGAIL w/ domain confusion loss?
- why does the paper report the "max cumulative reward so far" not the expected cumulative reward of the current episode? the latter could give a better idea of the training stability of the different algorithms
- out of curiosity: the shown scenarios are visually still pretty similar between source and target (eg background etc) --> how do you think an approach like the proposed one would scale to visually substantially different environments, eg from one kitchen to another?

## Suggestions to improve the paper
- add an additional baseline "DisentanGAIL w/ domain confusion loss & prior data", as discussed in the "weaknesses" section, particularly for the transfer tasks on the bottom right of Fig 2 in which the discrepancy between DisentanGAIL and the baselines is the largest
- add evaluation of baselines (particularly DisentanGAIL w/ domain confusion loss (w/ and w/o prior data)) to the harder manipulation environments in Fig 3 to show the benefits of the introduced regularizations
- since the proposed method addresses a concrete problem of prior work (as explained in appendix Section A) with a clear intuition, it could be nice to add a toy experiment early in the paper that demonstrates this effect empirically for an easy-to-analyze imitation problem, showing that for MI=0 the agent cannot properly learn to imitate since it is unable to capture the relevant information --> such an experiment could help to further motivate the need for the new regularizations
- the additional assumption of a "prior dataset" collected in both domains for additionally constraining the latent representation is first mentioned in section 4.2 --> since this is a different assumption from prior work on cross-domain imitation it would be good to mention this earlier, maybe in a dedicated "Problem Statement" section
- for qualitative matching results in Fig 4 in the appendix it would be nice to show the corresponding matches found when using the domain confusion loss instead of the proposed regularizations to see whether some of the failure cases are interpretable
- I wonder whether it would be possible to show imitation across agents with more drastic morphology differences in the most challenging 7DOF robotic manipulation tasks. Right now, while there are differences in the link dimensions, the main difference still seems to be in the visual appearance (at least from looking at the provided pictures) --> maybe one could try to eg try to transfer between robots with different gripper morphologies (eg one agent has U-shaped gripper vs another agent with T-shaped "gripper" for the pusher task), or from a 4DOF arm to a 7DOF arm etc.
- as mentioned in one question above: while the agent appearance and morphology changes between the experiments, the largest part of the observation, the background, is usually constant across source and target environment --> it would be interesting to see experiments with different background appearances to see the robustness of the method

## Overall Recommendation
The paper is well written and the experiments are covering a wide array of third-person visual imitation problems on which the proposed method shows strong results. The experimental evaluation seems thorough, all design choices are ablated. My main doubts are about the fairness of comparison to the baselines (particularly with regard to the additional data available to the proposed method), and some lacking baseline results on the harder tasks. Therefore I cannot fully support acceptance yet, but if the authors are able to provide the additional evaluations and answer the questions posed above adequately, I am willing to increase my score and vote for acceptance.


## Post-Rebuttal Comments
I thank the authors for their detailed feedback. Particularly the clarifications about the usage of prior data in the baselines were very helpful and the added results with background differences are interesting!

I am not sure whether it is standard to show the learning curves with the "max achieved return so far" in the GAIL literature, but if not I still think the true reward per episode should be shown to properly reflect the training stability of the algorithm. Potentially, the stability could also be increased by adding a learning rate decay schedule?

Overall, the rebuttal addressed my questions and incorporated many of the suggestions, therefore I am increasing my score and vote for acceptance.

---

> ### Author Response · Authors · 2020-11-21
> **Responses to AnonReviewer3 - part 1**
>
> **Weaknesses**
>
> 1. not fully fair comparison to baselines: since the main novelty lies in the introduction of novel regularization objectives, the "DisentanGAIL w/ domain confusion loss" is the main comparison method since the only difference to the proposed method is the representation regularization function. However, this baseline does not have access to the additional collected prior datasets used for the introduced "prior regularization" loss. Looking at Figure 2, it seems that the performance of DisentanGAIL w/o Prior Data and DisentanGAIL w/ domain confusion loss are near-equivalent which suggests that the access to the additional prior data might be the main factor that contributes to DisentanGAIL's superior performance --> an additional baseline "DisentanGAIL w/ domain confusion loss & prior data" is needed, which additionally trains the domain confusion objective on the prior data collected for the DisentanGAIL prior regularization objective, to allow for fair comparison of both regularization approaches with access to the same data
>
> We understand the reviewer's concerns and we would like to clarify that the algorithms making use of the domain confusion loss (TPIL and DisentanGAIL with DCL) do make use of prior data as additional training inputs with class label 0, and the correct domain label. This is analogous to the way the original TPIL algorithm made use of 'failure data', sampled by failure policies in both expert and agent domains. Enforcing only the expert demonstration constraint outperforms the domain confusion loss on most examined problems, even if this version of DisentanGAIL does not have access to prior data. Following the reviewer's comments, we realize how this information might have been unclear and have therefore added it explicitly in the latest revision (see Appendix B.1).
>
> 2. missing baseline results on harder tasks: on the harder tasks shown in Fig 3 there is no evaluation of the baseline methods, which makes it hard to judge how hard these tasks actually are for prior third-person visual imitation approaches
>
> In the latest revision, we collected the results and added the relative performance curves for all the remaining baselines in the high dimensional observational imitation problems. We also added Table 2 in Section 6 summarizing these results. We did not originally include them because some algorithms, such as TPIL, yielded very suboptimal results in preliminary experiments. Therefore, we believed their inclusion would provide little information.
>
> **Questions**
>
> 1. from my understanding the issue of prior work that constrains the domain-related MI to 0 (described in Appendix, section A) appears for any two domains (since, particularly in the beginning of training, there will always be differences in the goal-reaching distributions of expert and policy data). However, it seems that DisentanGAIL w/ domain confusion loss, which applies the MI=0 loss from Stadie et al. 2017, works well on many of the tested domains. How does that fit to the explanation made in Section A + the quant results in Fig 5?
>
> While Stadie et al. aim to constrain the mutual information of some latent representation with the domain labels to 0, in practice, their algorithm does not try to enforce this precisely. Particularly, the domain confusion loss with a fixed weight coefficient acts only as a heuristic penalty related to some quantity proportional to the mutual information contained in the latent representations. In the current revision, we expanded our discussion about these implications in Appendix A.
>
> 2. what differences result in the substantial performance difference between TPIL and DisentanGAIL w/ domain confusion loss?
>
> While DisentanGAIL with domain confusion loss shares most of the implementation details with DisentanGAIL (described in Appendix B), we obtained the TPIL results by adapting the authors' original code. The main distinctive features of DisentanGAIL with domain confusion loss, which likely had a significant effect on performance are:
> - Off-policy maximum-entropy actor - enabling more sample-efficient learning than TRPO and incentivizing the agent to explore even when an uninformative learning signal is provided.
> - Discriminator architecture and processing of the pre-processor's stochastic output (as described in Appendix B) - providing some implicit regularization and permitting to easily diminish the information contained in any of the independent dimensions of the Gaussian representations.
> - Invariant discriminator regularization - providing further regularization for the expressivity of the invariant discriminator.

---

> > ### Author Response · Authors · 2020-11-21
> > **Responses to AnonReviewer3 - part 2**
> >
> > **Questions**
> >
> > 3. why does the paper report the "max cumulative reward so far" not the expected cumulative reward of the current episode? the latter could give a better idea of the training stability of the different algorithms
> >
> > Once GAIL-based methods achieve optimality, the distribution of behavior in B_\pi and B_E become semantically similar. This causes instabilities since the discriminator still tries to perfectly discern the two distributions but cannot effectively use goal-completion information to do so. This made some of the performance curves harder to discern and, hence, harder to evaluate. Therefore, for visual clarity, we opted to display the described metric.
> >
> > 4. out of curiosity: the shown scenarios are visually still pretty similar between source and target (eg background etc) --> how do you think an approach like the proposed one would scale to visually substantially different environments, eg from one kitchen to another?
> >
> > Observational imitation is still a very much ill-posed problem and hence, we hypothesize that many of the recovered solutions are also possible thanks to the structural inductive biases of our model. Particularly, we believe convolutions play a big role in guiding the optimization towards looking for locally consistent shared elements across the expert and agent observations. Preliminary unsuccessful experiments in environments with few correspondences between the agent and expert domains (e.g., different objects to be moved to different targets on different tables) seemed to be in line with this hypothesis. Hence, moving forward, we believe data augmentation and meta-learning can play a key role in combating some of these issues. Particularly, data-augmentation can help the model understand which characteristics of the observation are not a consequence of the expert's behavior and should therefore be disregarded. Similarly, meta-learning could help the model build a prior over which features contain 'plausible' information about the goal completion levels within a distribution of observational imitation problems.
> >
> >
> > **Suggestions**
> >
> > 1. add an additional baseline "DisentanGAIL w/ domain confusion loss & prior data", as discussed in the "weaknesses" section...
> >
> > (From answer to Weaknesses 1) We would like to clarify that the algorithms making use of the domain confusion loss (TPIL and DisentanGAIL with DCL) do make use of prior data as additional training inputs with class label 0, and the correct domain label. This is analogous to the way the original TPIL algorithm made use of 'failure data', sampled by failure policies in both expert and agent domains. Enforcing only the expert demonstration constraint outperforms the domain confusion loss on most examined problems, even as this version of DisentanGAIL does not have access to prior data. Following the reviewer's comments, we realize how this information might have been unclear and added it explicitly in the latest revision (see Appendix B.1).
> >
> > 2. add evaluation of baselines (particularly DisentanGAIL w/ domain confusion loss (w/ and w/o prior data)) to the harder manipulation environments in Fig 3 to show the benefits of the introduced regularizations
> >
> > (From answer to Weaknesses 2) We collected the results and added the relative performance curves for all the remaining baselines in the high dimensional observational imitation problems. We also added Table 2 in Section 6 summarizing these results.
> >
> > 3. since the proposed method addresses a concrete problem of prior work (as explained in appendix Section A) with a clear intuition, it could be nice to add a toy experiment early in the paper that demonstrates this effect empirically for an easy-to-analyze imitation problem, showing that for MI=0 the agent cannot properly learn to imitate since it is unable to capture the relevant information --> such an experiment could help to further motivate the need for the new regularizations
> >
> > Such an experiment could surely solidify the proposed arguments for aiming to enforce a different constraint. However, we believe that making a stable algorithm which truly enforces MI=0 is not a trivial task. In particular, both a domain confusion loss with a very high weighting coefficient and a dual penalty with I_{max}=0 might lead to some minimal domain information leaking into the latent representations, simply due to the stochastic nature of the training process. On the other hand, we think that the empirical issues of enforcing a low mutual information constraint are also somewhat shown by the experiments in section D.3 of the Appendix. Here, setting I_{max}=0.01 for the expert demonstration constraint leads to a consistent reduction in performance in any experiment considering domain differences. We expanded Section 4.2 in the main text and section A in the Appendix to more thoroughly discuss the issues related to the constraint proposed by Stadie et al. and the effects of the domain confusion loss.

---

> > > ### Author Response · Authors · 2020-11-21
> > > **Responses to AnonReviewer3 - part 3**
> > >
> > > **Suggestions**
> > >
> > > 4. the additional assumption of a "prior dataset" collected in both domains for additionally constraining the latent representation is first mentioned in section 4.2 --> since this is a different assumption from prior work on cross-domain imitation it would be good to mention this earlier, maybe in a dedicated "Problem Statement" section
> > >
> > > We have now added an introductory paragraph in Section 4 that introduces DisentanGAIL and defines prior data before delving into the algorithm's details.
> > >
> > > 5. for qualitative matching results in Fig 4 in the appendix it would be nice to show the corresponding matches found when using the domain confusion loss instead of the proposed regularizations to see whether some of the failure cases are interpretable
> > >
> > > In the latest revision, we have expanded section D.1 to report additional results showing the observations couplings produced by DisentanGAIL with domain confusion loss.
> > >
> > > 6. I wonder whether it would be possible to show imitation across agents with more drastic morphology differences in the most challenging 7DOF robotic manipulation tasks. Right now, while there are differences in the link dimensions, the main difference still seems to be in the visual appearance (at least from looking at the provided pictures) --> maybe one could try to eg try to transfer between robots with different gripper morphologies (eg one agent has U-shaped gripper vs another agent with T-shaped "gripper" for the pusher task), or from a 4DOF arm to a 7DOF arm etc.
> > >
> > > We realize that we originally omitted from Appendix C the fact that the gripper morphologies do differ in the Pusher task (the Human-like 7DOF-Pusher has a 4-linked thick gripper while the original 7DOF-Pusher has a 3-linked thin gripper). We have now included this information in the current revision. The visualizations in Appendix D.1, regarding the manipulation tasks, indicate that DisentanGAIL effectively learns to encode information about the objects’ positions, which should be unaffected by the agents’ morphology differences. Therefore, we hypothesize that observational imitation between increasingly different agents should be possible. However, DisentanGAIL would have fewer meaningful correspondences to learn the appropriate motion to move the object towards the target, making the initial exploration stage harder.
> > >
> > > 7. as mentioned in one question above: while the agent appearance and morphology changes between the experiments, the largest part of the observation, the background, is usually constant across source and target environment --> it would be interesting to see experiments with different background appearances to see the robustness of the method
> > >
> > > Following the reviewer's suggestion, in the latest revision we added section D.4 to the Appendix, where we test DisentanGAIL on two alternative target environments in the Hopper and 7DOF-Pusher environment realms, with very distinct backgrounds from the relative source environments:
> > > -	the new target environment in the Hopper realm has a much darker floor, where the tiles are difficult to discern.
> > > -	the new target environment in the 7DOF-Pusher realm includes a green table and a white floor, both of which differ greatly in appearance to the grey table and black floor of the source environment.
> > >
> > > The obtained results show that background domain differences have a minor effect on DisentanGAIL’s final performance. However, they have a more prominent effect on DisentanGAIL’s efficiency, making our algorithm converge in an increased number of epochs. This is particularly noticeable in the Hopper realm’s results. We hypothesize this is because in these experiments there is a greater amount of domain information that requires to be ‘disentangled’ from the useful goal-completion information. For example, in the locomotion realms, the pre-processor encodes goal-completion information about the relative position of the agent with respect to the floor tiles in the two environments (as empirically suggested in Section D.1). In the new target environment of the Hopper realm, this information needs to be also disentangled from domain information regarding the tiles’ appearance.

---

### Official Review · AnonReviewer1 · 2020-10-29

**Rating:** 7
**Confidence:** 3

**Review:**

Summary
--
This paper proposes a method for performing observational imitation learning -- an existing task that seeks to enable an agent to learn from visual observations of expert behavior in order to roughly imitate the expert behavior. The method employs an adversarial imitation learning objective function that incorporates proposed mutual information constraints that are intended to force the representation space to be invariant to the domain of the data sources, and instead only encode goal-completion information.

Experiments illustrate that the method was able to learn and be performant despite visual and embodiment differences in the expert and agent domain on various mujoco environments, exhibiting substantially better performance to a different observational IL method. Experiments also illustrate good performance on a slightly higher dimensional task.

Clarity and Correctness
--
The paper has a few somewhat painful clarity issues that make it difficult to understand.

- Section 4 starts with a subsection that describes components of the architecture, when instead it should start with a clear high-level description of the approach.

- S4.2 The relationship between B_P.E and B_P.\pi and B_E and B_\pi is unclear. The former two terms were never defined in the preliminaries, so it's not clear whether they constitute additional data that was not mentioned in the assumptions. This ties into my comment about about Section 4 needing an introductory paragraph to lay out the high-level assumptions, inputs and outputs, and idea of the method. The notion of "prior data" is used in the paper, but I cannot find a clear description of it anywhere.

Originality
--
As mentioned in the paper, this paper is closely related to Stadie et al. (2017). There's description of the differences to it in various places throughout the paper, but it would be nice if there were a clear section on the comparison of objective functions.

Significance
--
The significance of this paper is that it demonstrates a more performant method for performing observational imitation learning, which is potentially more applicable than standard IL approaches that require state or state-action traces.

- Why was TPIL not used as a method of comparison in Fig 3?

- The results shown in Fig 2 have rather short x-axes, with a maximum of 20,000 steps of training. The comparisons would be more informative if training were run for more steps (e.g. 1e6 or 1e7). It is possible that the other approaches end up matching expert performance as well, without any great loss in efficiency.

Other comments
--
- S4.2 prior data constraint -- the inequality is backwards (MI is nonnegative)
- S3.2, trajectories are sequences, use () not {}
- The logical and notation in S4.2 is very confusing -- the statements inside the parentheses that define d_i aren't boolean truth values, so it makes the definition of d_i confusion. My suggestion is just to use d_i = 1(o \in B_e), since B_e and B_\pi are disjoint.
- Be precise / define what you mean by "high dimensional". "High" is subjective, and is perhaps not the most appropriate adjective to describe 7 dimensional tasks. Also, make it clear that the high dimensionality is in the action space.

Post-rebuttal comments
--
After reading the authors' response and the updated components of the manuscript, I thank the authors for addressing nearly all of my concerns. The inclusion of a clearer motivation, more discussion w.r.t. TPIL, and comparison to TPIL, all enhance my understanding of the contributions of the paper beyond my original review enough for me to increase my score from a 6 to a 7.

---

> ### Author Response · Authors · 2020-11-21
> **Responses to AnonReviewer1**
>
> **Clarity and Correctness**
>
> 1. Section 4 starts with a subsection that describes components of the architecture, when instead it should start with a clear high-level description of the approach.
> 2. The relationship between B_{P.E} and B_{P.\pi} and B_E and B_\pi is unclear. The former two terms were never defined in the preliminaries, so it's not clear whether they constitute additional data that was not mentioned in the assumptions. This ties into my comment about Section 4 needing an introductory paragraph to lay out the high-level assumptions, inputs and outputs, and idea of the method. The notion of "prior data" is used in the paper, but I cannot find a clear description of it anywhere.
>
> In the current revision, we added a paragraph at the beginning of Section 4 that provides a high-level overview and explains the objectives of DisentanGAIL, referencing the previously introduced notation. In the same paragraph, we also describe the nature of prior data as representing unsupervisedly-collected observations of both expert’s and agent’s domains and define its notation.
>
> **Originality**
>
> 1. As mentioned in the paper, this paper is closely related to Stadie et al. (2017). There's description of the differences to it in various places throughout the paper, but it would be nice if there were a clear section on the comparison of objective functions.
>
> We extended Section 4.2 with a paragraph discussing the implications of the constraint from Stadie et al. and its implementation, as compared to ours.
>
>
> **Significance**
>
> 1. Why was TPIL not used as a method of comparison in Fig 3?
>
> In the latest revision, we collected the results and added the relative performance curves for all the remaining baselines in the high dimensional observational imitation problems. We also added Table 2 in Section 6 summarizing these results. We did not originally include these results because some algorithms, such as TPIL, yielded very suboptimal results in preliminary experiments. Therefore, we believed their inclusion would provide little information.
>
> 2. The results shown in Fig 2 have rather short x-axes, with a maximum of 20,000 steps of training. The comparisons would be more informative if training were run for more steps (e.g. 1e6 or 1e7). It is possible that the other approaches end up matching expert performance as well, without any great loss in efficiency.
>
> While we agree that running for a much greater number of steps would be more informative, in the low dimensional experiments we wanted to focus on providing a wide range of results, testing a multitude of methods and domain differences. This would not have been possible using our computational resources if we collected orders of magnitude of additional training steps per experiment. Moreover, one of the goals of our experiments was evaluating the algorithms for their efficiency, in order to understand their potential real-world applicability. Hence, in this context, we believe that 20,000 frames of experience are a fair allowance for solving the low dimensional task.
>
> **Other comments**
>
> We addressed the reviewer’s additional comments in the latest version of the paper. Particularly, we fixed the inexact notation, changed the definition of d_i and explicitly defined the notion of 'high dimensional tasks' in Section 6, as suggested.

---

### Official Review · AnonReviewer2 · 2020-10-30
**Review #2**

**Rating:** 6
**Confidence:** 4

**Review:**

This paper proposes a visual imitation learning algorithm that can handle domain shifts between the expert demonstrations and the data generated by the agent. The proposed method is built upon GAIL and scales to image inputs. The authors handle the domain shift problem by learning a domain-invariant discriminator and a statistics network. The invariant discriminator takes in a concatenation of a sequence of latent representations of the observations and tries to predict without relying on the domain information. The statistics network is used for estimating the mutual information constraint between the latent representation and the domain labels, which the authors are trying to minimize in order to attain domain invariant predictions. The authors optimize the standard GAIL objectives along with the mutual information constraints as regularizations jointly. The experiments are done in both discrete and continuous control tasks in Mujoco and the proposed approach, DisentanGAIL, outperforms the prior method TPIL.\

For pros, I think the experimental setup is reasonable and the authors conduct relatively thorough ablation studies to show the usefulness of various components such as the prior data constraint, double statistics network, and spectral normalization regularization, which is helpful for understanding the method. The paper is also well written and easy to understand.

However, I have a few concerns about this approach, which I will list as follows.

1. Regarding the novelty of this paper, I feel like the approach is not particularly novel. The method is similar to TPIL (Stadie et al. 2017) and the main difference between DisentanGAIL and TPIL is that in DisentanGAIL, the authors set the mutual information constraint to be less than 1 bit, and in TPIL, the mutual information is constrained to 0. I can definitely see why the softer regularization works better, but I wonder if the contribution of this work is substantial given such a simple tweak. Moreover, the authors also propose many techniques to make DisentanGAIL work better, such as using unsupervised data with some regularization, double statistics networks, spectral norm, and etc.. However, these seem to be more like implementation tricks rather than major contributions and I'm unsure if they make the contribution substantial enough for the standard of an ICLR paper. Moreover, adding these additional components definitely seem to make the approach much more complex and might require much more tuning to get it to work, which is another concern.

2. Another point related to novelty and related works, there are several papers that also consider domain-adaptive imitation learning that seem pretty similar to this paper, such as [1, 2, 3]. [1, 2] consider learning domain-invariant features, while [3] proposes a unifying theoretical framework for domain-adaptive imitation learning. I think these methods could serve as comparisons to DisentanGAIL.

3. Furthermore, since the approach seems to be a bit incremental, it would be nice to have the theoretical analysis that could justify the incremental changes and guarantee the convergence to the optimal policy (e.g. attaining similar results in [3]). Unfortunately, this is missing in the paper.

4. The challenging, high-dimensional environments used in the paper are all in locomotion tasks in Mujoco. It would be nice to see more realistic environments such as robotic manipulation tasks like ROBEL and  Adroit etc., which would make the domain adaptation more appealing.

Overall, based on the arguments above, I would recommend a reject for this paper.

------------------------------------------------------------------------------------------------------------------
Post-rebuttal updates:

After reading the author response and other reviews, I agree that the difference between the paper and prior works is now much more clear and the empirical evidence shows that the new method works well, though I'm still concerned about the part where the authors add many components and make the algorithm much more complex and potentially hard to work in practice. Nevertheless, I've increased my score to a 6.

[1] Okumura, Ryo, Masashi Okada, and Tadahiro Taniguchi. "Domain-Adversarial and-Conditional State Space Model for Imitation Learning." arXiv preprint arXiv:2001.11628 (2020).
[2] Lu, Yiren, and Jonathan Tompson. "ADAIL: Adaptive Adversarial Imitation Learning." arXiv preprint arXiv:2008.12647 (2020).
[3] Kim, Kuno, et al. "Domain adaptive imitation learning." arXiv preprint arXiv:1910.00105 (2019).

---

> ### Author Response · Authors · 2020-11-21
> **Responses to AnonReviewer2 - part 1**
>
> **Concerns:**
>
> 1. Regarding the novelty of this paper, I feel like the approach is not particularly novel. The method is similar to TPIL (Stadie et al. 2017) and the main difference between DisentanGAIL and TPIL is that in DisentanGAIL, the authors set the mutual information constraint to be less than 1 bit, and in TPIL, the mutual information is constrained to 0. I can definitely see why the softer regularization works better, but I wonder if the contribution of this work is substantial given such a simple tweak. Moreover, the authors also propose many techniques to make DisentanGAIL work better, such as using unsupervised data with some regularization, double statistics networks, spectral norm, and etc.. However, these seem to be more like implementation tricks rather than major contributions and I'm unsure if they make the contribution substantial enough for the standard of an ICLR paper. Moreover, adding these additional components definitely seem to make the approach much more complex and might require much more tuning to get it to work, which is another concern.
>
> There are several core differences between DisentanGAIL and prior works, which make our approach concretely novel. In particular, while Stadie et al. argue for constraining the mutual information of some representation with the domain labels to be 0, in practice they try to enforce this very loosely via an additional domain confusion loss with a fixed weight coefficient. This is very different, in both aims and implementation, from our approach of calculating the mutual information explicitly and applying the adaptive and dual penalization terms to precisely ensure the enforcement of the proposed constraints. Moreover, we argue that the identification of the domain information disguising phenomenon is also a valuable contribution to solve the problem of observational imitation. In this regard, the proposed double statistics network and invariant discriminator regularization can become standard practice for future efforts to ensure its prevention. While we understand the reviewer's concern about the increased complexity, we would like to emphasize that all the above-mentioned practices did not add any hyper-parameter which required tuning across our diverse set of experiments. Thus, we do not think this should hinder the applicability of our algorithm. We believe our contribution beyond prior works is also substantiated by our empirical results. Our algorithm can be successfully applied to high dimensional environments, under limited experience regimes, even for problems considering major domain appearance and embodiment differences. In contrast, to the best of our knowledge, prior works showed algorithms based on the domain confusion loss to be successful solely in low dimensional environments (where agents had maximum 2 degrees of freedom) for problems only considering domain appearance differences. Taking into account the reviewer's concerns, we extended section 4.2 in the main text and section A in the Appendix to more thoroughly discuss the work from Stadie et al. and its differences with ours.
>
> 2. Another point related to novelty and related works, there are several papers that also consider domain-adaptive imitation learning that seem pretty similar to this paper, such as [1, 2, 3]. [1, 2] consider learning domain-invariant features, while [3] proposes a unifying theoretical framework for domain-adaptive imitation learning. I think these methods could serve as comparisons to DisentanGAIL.
>
> We thank the reviewer for bringing to our attention DAC-SSM [1] from Okumura et al. In this work, the authors combine the domain confusion loss from Stadie et al. with a modified optimization scheme and a model-based agent. However, as in the original Third-Person Imitation Learning paper, the authors show successful applications exclusively in low dimensional environments considering only domain appearance differences. We now added a reference of DAC-SSM in the Related work section. We are not able to provide a direct comparison as the authors do not provide a publicly accessible implementation of their algorithm or their environments.
>
> In the Related work section, we did already mention the connection of our work with DAIL [3] from Kim et al. In particular, this work assumes access to aligned demonstrations from environments where both agent and expert already achieved expertise, in both the agent's and the expert's point of view. In contrast, we do not make such an assumption and tackle a similar but more general problem-setting.
>
> ADAIL [3], from Lu et al., tackles a problem which can be described as agent-centric imitation under differences in the expert's and agent's domains dynamics. While this problem is surely related, we believe it to be too distant from the problem of observational imitation tackled in our work.

---

> > ### Author Response · Authors · 2020-11-21
> > **Responses to AnonReviewer2 - part 2**
> >
> > 3. Furthermore, since the approach seems to be a bit incremental, it would be nice to have the theoretical analysis that could justify the incremental changes and guarantee the convergence to the optimal policy (e.g. attaining similar results in [3]). Unfortunately, this is missing in the paper.
> >
> > We argue that our work makes already a significant contribution towards solving the problem of observational imitation, as highlighted in our responses above. In their derivations, Kim et al. consider a limited class of alignable MDPs and only show that a mapping between optimal policies in this scenario should exist. On the other hand, imitation across non-alignable POMDPs is a majorly ill-posed problem and making rigorous claims would require major assumptions about the tasks and the expert demonstrations. While we understand the importance of such analysis, we leave this to future works focusing on more restricted classes of observational imitation problems, in which such assumptions can apply.
> >
> > 4. The challenging, high-dimensional environments used in the paper are all in locomotion tasks in Mujoco. It would be nice to see more realistic environments such as robotic manipulation tasks like ROBEL and Adroit etc., which would make the domain adaptation more appealing.
> >
> > Our high dimensional environments do not solely involve locomotion tasks but also two robotic manipulation tasks, namely within the 7DOF-Pusher and 7DOF-Striker environment realms. Particularly, these manipulation tasks examine observational imitation from a human-like agent to a robot-like agent based on the PR2, with major domain differences in terms of both agent appearance and embodiment (as described in Appendix C). We have released our environment suite as part of our submission, which already provides a more challenging and diverse set of observational imitation problems than what was previously attempted by prior related works. However, we agree with the reviewer regarding the importance of even more realistic and challenging problems and we hope that future works will extend our efforts and add additional realistic robotics manipulation problems to this suite.

---

### Official Review · AnonReviewer4 · 2020-11-01
**Learning domain-invariant representations for observational imitation learning**

**Rating:** 7
**Confidence:** 3

**Review:**

This paper studies observational imitation learning, a problem setting in which the agent wishes to learn from expert observations, but the state, action, and observational spaces of the expert and agent domains can vary. They introduce DisentanGAIL to tackle this problem by introducing two mutual information constraints in the GAIL framework to learn domain-invariant representations of observations. In particular, DisentanGAIL aims to discard domain information in the presentation, while retaining relevant goal completion information.

Overall, the problem setting is very practical. The performance gain is greatest in the domains where the camera angle changes, which is promising. I like the intuition provided in Appendix A and think some of it should be included in the main text. It motivates well the distinction between domain information and goal-completion, and highlights why previous methods like domain confusion loss can fail. Moreover, I would’ve liked to see some analysis of the experiments to support this hypothesis. For example, a version of Appendix D1 would be valuable to include, and perhaps also another visualization demonstrating that other baselines fail to encode goal completion information. Some of the implementation details in Section 5 can be shortened/moved to the appendix instead.

Some questions:

- What is the difference between DisentanGAIL with DCL and TPIL? The text says that the DCL substitutes the mutual information constraints, so the only difference I see would be learning a distribution over latent representations rather than a deterministic feature extractor. Any intuition why this alone leads to such a big difference in Table 1?

- Without any regularization, I’m surprised DisentanGAIL learns anything at all. Wouldn’t the discriminator then be able to tell the two domains apart (especially if one of the differences is color) and fail to provide a meaningful learning signal to the agent?

- If I understand Figure 3 correctly, the orange line is without any regularization (lower bound) and the green is learning without domain differences (upper bound). How come DisentanGAIL can eventually outperform the green line in some of the experiments?

- What’s the gap between DisentanGAIL and the expert policy? Does the expert always achieve a reward of 1?

Other comments

- Please clean up all Figures and Tables. The text is too small and impossible to read without zooming in very close.

- “Learning” typo in last line of page 2.

----------

Post-rebuttal comments

Thank you for answering my all questions and updating the manuscript with some of my suggestions. The additions in section 4 clarify the motivation much better and also highlight the differences with prior work. I've increased my score from 6 to 7.

---

> ### Author Response · Authors · 2020-11-21
> **Responses to AnonReviewer4**
>
> **General comments:**
>
> 1. I like the intuition provided in Appendix A and think some of it should be included in the main text.
>
> In the current revision, we extended Section 4.2 in the main text to include an explanation of the downsides of the constraint proposed by Stadie et al. We also expanded Appendix A to give further details of the implications of their practical implementation.
>
> 2. a version of Appendix D1 would be valuable to include, and perhaps also another visualization demonstrating that other baselines fail to encode goal completion information.
>
> In the current revision, we expanded Appendix D.1 with an additional visualization, showing the observation couplings produced by DisentanGAIL with DCL. Due to space constraints, we were not able to include a version of this section in the main text and we opted to keep it as part of the Appendix.
>
> **Questions:**
>
> 1. What is the difference between DisentanGAIL with DCL and TPIL?
>
> While DisentanGAIL with DCL shares most of the implementation details with DisentanGAIL (described in Appendix B), we obtained the TPIL results by adapting the authors' original code. The main distinctive features of DisentanGAIL with DCL, which likely had a significant effect on performance are:
> - Off-policy maximum-entropy actor - enabling more sample-efficient learning than TRPO and incentivizing the agent to explore even when an uninformative learning signal is provided.
> - Discriminator architecture and processing of the pre-processor's stochastic output (as described in Appendix B) - providing some implicit regularization and permitting to easily diminish the information contained in any of the independent dimensions of the Gaussian representations.
> - Invariant discriminator regularization - providing further regularization for the expressivity of the invariant discriminator.
>
> 2. Without any regularization, I’m surprised DisentanGAIL learns anything at all. Wouldn’t the discriminator then be able to tell the two domains apart (especially if one of the differences is color) and fail to provide a meaningful learning signal to the agent?
>
> Even without any regularization, DisentanGAIL still appears to provide some meaningful learning signal to the actor, outperforming TPIL on many problems. We believe that this is possible thanks to the same components allowing DisentanGAIL with DCL to substantially outperform TPIL. For example, the Gaussian representation of the pre-processor's output should always provide some soft regularization. When this is combined with the inherent stochasticity of the training process, it can lead to the invariant discriminator to make use of some useful goal-completion information while providing rewards to the agent's transitions. However, in most examined problems, these components alone are not enough to effectively reproduce the demonstrated task, as highlighted by the significant performance gap with the other DisentanGAIL ablations.
>
> 3. If I understand Figure 3 correctly, the orange line is without any regularization (lower bound) and the green is learning without domain differences (upper bound). How come DisentanGAIL can eventually outperform the green line in some of the experiments?
>
> The environments tested in the high dimensional experiments pose challenges even in the setting of imitation without domain differences. For the manipulation tasks, we observed that the proposed mutual information constraints (which the algorithm learning without domain differences does not apply) are actually beneficial regardless of the presence of domain differences. We hypothesize that this is due to the general regularizing effect that they have on the discriminator, counteracting the manipulation environments' inherent stochasticity. Particularly, the expert data constraint has the side effect of limiting information flow in the discriminator (similarly to Peng et al., 2018), while the prior data constraint can help to identify which information is irrelevant for achieving the demonstrated objective (similarly to Zolna et al., 2019). We hypothesize that these factors allow the full DisentanGAIL algorithm to slightly exceed the performance of the source environment baseline in the 7DOF-Striker realm.
>
> 4. What’s the gap between DisentanGAIL and the expert policy? Does the expert always achieve a reward of 1?
>
> We normalize the reported performance such that 1 represents the episode returns achieved by the expert agent. DisentanGAIL always recovers a comparable performance to the expert policy, with a larger gap in the high dimensional locomotion environments (as discussed in Section 6).
>
> **Other comments**
>
> 1. Please clean up all Figures and Tables. The text is too small and impossible to read without zooming in very close.
>
> In the current revision, we have increased the size of the figures and tables, together with their corresponding text.
>
> 2. “Learning” typo in last line of page 2.
>
> Fixed.

---

> > ### Author Response · Authors · 2020-11-21
> > **References**
> >
> > Bradly C Stadie, Pieter Abbeel, and Ilya Sutskever. Third-person imitation learning. arXiv preprint
> > arXiv:1703.01703, 2017.
> >
> > Xue Bin Peng, Angjoo Kanazawa, Sam Toyer, Pieter Abbeel, and Sergey Levine. Variational discriminator
> > bottleneck: Improving imitation learning, inverse rl, and gans by constraining information
> > flow. arXiv preprint arXiv:1810.00821, 2018.
> >
> > Konrad Zolna, Scott Reed, Alexander Novikov, Sergio Gomez Colmenarej, David Budden, Serkan
> > Cabi, Misha Denil, Nando de Freitas, and Ziyu Wang. Task-relevant adversarial imitation learning.
> > arXiv preprint arXiv:1910.01077, 2019.

---

### Author Response · Authors · 2020-11-21
**Update overview**

We would like to truly thank all the reviewers for their constructive feedback and impactful suggestions. We have incorporated many of these suggestions into our paper, helping us considerably improve the completeness and readability of this work. We have posted responses to the individual reviews, addressing each question and concern in detail. Below, we provide an overview of the major changes we have made in the current revision of our paper:

•	We added an introductory paragraph to Section 4 which gives a high-level overview of DisentanGAIL and defines the notion of prior data.

•	We extended Section 4.2  by including some of the intuition from Appendix A, to provide a comparison of our methodology with the constraint proposed by Stadie et al. and its implementation.

•	We specified the criterion used to characterize the Hopper, Half-Cheetah, 7DOF-Pusher and 7DOF-Striker environments as 'high dimensional' in Section 6.

•	We collected the results and added the relative performance curves for all the remaining baselines in the high dimensional observational imitation problems. We also added Table 2 in Section 6 summarizing these results.

•	We extended Appendix A by further discussing practical implications of the domain confusion loss proposed by Stadie et al.

•	We clarified how and when prior data is used by the different ablations in Appendix B.1.

•	We expanded section D.1 of the Appendix to report additional results showing the latent representations couplings produced by DisentanGAIL with domain confusion loss.

•	We added Section D.4 in the Appendix, where we evaluate DisentanGAIL on two alternative target environments in the Hopper and 7DOF-Pusher realms with very distinct backgrounds from the relative source environments.

---

### Decision · Program_Chairs · 2021-01-07
**Final Decision**

**Decision:**

Accept (Poster)

**Comment:**

The reviewers raised a number of concerns about the novelty of the paper and comparisons. The authors were able to address the concerns regarding the comparisons in the response, and the reviewers unanimously agree that the paper should be published. I do think however that this paper is quite borderline. I agree with the reviewers that the updated experiments are convincing in terms of the provided comparisons. However, the reservations I have about the work can perhaps best be stated as follows: There is quite a bit of work in the area of imitation from observations, which makes a range of different assumptions and utilizes a variety of different domain adaptation techniques. Much of this work is in the robotics domain (which is cited in the paper), and much of it demonstrates results in fairly realistic settings, often with real humans and real robots. In comparison, the experiments in this paper are quite simplistic, using toy domains and "demonstrations" obtained from a computational oracle (i.e., another policy). Given the maturity of this field and the current state of the art, I am skeptical of this evaluation, and I think TPIL is a very weak baseline. That said, I would  defer to the reviewers in this case -- I do think the particular technical contributions that the paper makes are a valuable addition to the literature, though somewhat incremental. I am also sympathetic to the authors in that much of the more successful prior work in this area that does evaluate under realistic conditions makes subtly different assumptions, or utilizes different techniques for which it is difficult to provide an apples-to-apples comparison.

One thing I would request of the authors for the camera ready though is: Please tone down the claims. "Human-like 7 DOF Striker" is not human-like, it's a crudely simulated robotic arm that was recolored. It would of course be better to have a realistic evaluation (as many prior papers in this field indeed have), but in the absence of that, it is best not to overclaim and be upfront that the evaluation is on relatively simple simulated tasks under conditions that are not necessarily realistic (and have nothing to do with actual humans), but meant rather to evaluate in an apples-to-apples manner the particular algorithmic innovations in the method.